# Antioxidant, Cytotoxic, and Rheological Properties of Canola Oil Extract of *Usnea barbata* (L.) Weber ex F.H. Wigg from Călimani Mountains, Romania

**DOI:** 10.3390/plants11070854

**Published:** 2022-03-23

**Authors:** Violeta Popovici, Laura Bucur, Cerasela Elena Gîrd, Dan Rambu, Suzana Ioana Calcan, Elena Iulia Cucolea, Teodor Costache, Mădălina Ungureanu-Iuga, Mircea Oroian, Silvia Mironeasa, Verginica Schröder, Emma-Adriana Ozon, Dumitru Lupuliasa, Aureliana Caraiane, Victoria Badea

**Affiliations:** 1Department of Microbiology and Immunology, Faculty of Dental Medicine, Ovidius University of Constanta, 7 Ilarie Voronca Street, 900684 Constanta, Romania; violeta.popovici@365.univ-ovidius.ro (V.P.); victoria.badea@365.univ-ovidius.ro (V.B.); 2Department of Pharmacognosy, Faculty of Pharmacy, Ovidius University of Constanta, 6 Capitan Al. Serbanescu Street, 900001 Constanta, Romania; 3Department of Pharmacognosy, Phytochemistry and Phytotherapy, Faculty of Pharmacy, Carol Davila University of Medicine and Pharmacy, 6 Traian Vuia Street, 020956 Bucharest, Romania; 4Research Center for Instrumental Analysis SCIENT, 1E Petre Ispirescu Street, 077167 Tâncăbești, Romania; dan.rambu@scient.ro (D.R.); suzana.calcan@scient.ro (S.I.C.); iulia.cucolea@scient.ro (E.I.C.); teodor.costache@scient.ro (T.C.); 5Faculty of Food Engineering, Stefan cel Mare University of Suceava, 13th University Street, 720229 Suceava, Romania; silviam@fia.usv.ro; 6Integrated Center for Research, Development, and Innovation in Advanced Materials, Nanotechnologies and Distributed Systems for Fabrication and Control (MANSiD), Stefan cel Mare University of Suceava, 13th University Street, 720229 Suceava, Romania; 7Department of Cellular and Molecular Biology, Faculty of Pharmacy, Ovidius University of Constanta, 6 Capitan Al. Serbanescu Street, 900001 Constanta, Romania; 8Department of Pharmaceutical Technology and Biopharmacy, Carol Davila University of Medicine and Pharmacy, 6 Traian Vuia Street, 020956 Bucharest, Romania; dumitru.lupuliasa@umfcd.ro; 9Department of Oral Rehabilitation, Faculty of Dental Medicine, Ovidius University of Constanta, 7 Ilarie Voronca Street, 900684 Constanta, Romania; aureliana.caraiane@365.univ-ovidius.ro

**Keywords:** *Usnea barbata* (L.) Weber ex F.H. Wigg extract in canola oil, mineral content, usnic acid, phenolic compounds, antioxidant activity, cytotoxicity, BSL-test, rheological properties

## Abstract

*Usnea* genus (*Parmeliaceae*, lichenized *Ascomycetes*) is a potent phytomedicine, due to phenolic secondary metabolites, with various pharmacological effects. Therefore, our study aimed to explore the antioxidant, cytotoxic, and rheological properties of *Usnea barbata* (L.) Weber ex F.H. Wigg (*U. barbata*) extract in canola oil (UBO) compared to cold-pressed canola seed oil (CNO), as a green solvent used for lichen extraction, which has phytoconstituents. The antiradical activity (AA) of UBO and CNO was investigated using UV-Vis spectrophotometry. Their cytotoxicity was examined in vivo through a brine shrimp lethality (BSL) test after *Artemia salina* (*A. salina*) larvae exposure for 6 h to previously emulsified UBO and CNO. The rheological properties of both oil samples (flow behavior, thixotropy, and temperature-dependent viscosity variation) were comparatively analyzed. The obtained results showed that UBO (IC_50_ = 0.942 ± 0.004 mg/mL) had a higher 1,1-diphenyl-2-picrylhydrazyl (DPPH) radical scavenging activity than CNO (IC_50_ = 1.361 ± 0.008 mg/mL). Both UBO and CNO emulsions induced different and progressive morphological changes to *A. salina* larvae, incompatible with their survival; UBO cytotoxicity was higher than that of CNO. Finally, in the temperature range of 32–37 °C, the UBO and CNO viscosity and viscoelastic behavior indicated a clear weakening of the intermolecular bond when temperature increases, leading to a more liquid state, appropriate for possible pharmaceutical formulations. All quantified parameters were highly intercorrelated. Moreover, their significant correlation with trace/heavy minerals and phenolic compounds can be observed. All data obtained also suggest a possible synergism between lichen secondary metabolites, minerals, and canola oil phytoconstituents.

## 1. Introduction

In the plant world, the significant interest in lichens (lichenized fungi), symbionts between a fungus and one or more photosynthetic organisms (algae or cyanobacteria), is due to their specific secondary metabolites [1] that protect them against abiotic stress [2] and biological attack [3]. Numerous pharmacological studies proved their significant properties [4], the reason for lichen use in traditional medicine for thousands of years [5]. Nowadays, they are considered potential alternative remedies for modern diseases [6,7]. The most relevant lichen secondary metabolites are phenolic compounds; their reactive hydroxyl groups confer various bioactivities [8] through different mechanisms [4]. In addition to phenolic acids and tannins, lichens contain specific phenolic constituents, classified as depsides, depsidones, dibenzofurans, and pulvinic acid derivatives [9].

According to Prateeska et al. (2016) [10], in the lichen world, the genus *Usnea* (*Parmeliaceae*, lichenized *Ascomycetes*) can be considered a potent phytomedicine, with various pharmacological effects. Using UHPLC, high-resolution mass spectrometry (HRMS), and MS^2^ analysis, Salgado et al. (2018) [11] analyzed the secondary metabolites of *Usnea* lichen species. Thus, in a methanol extract of *U. barbata* (one of the best known representatives of *Usnea* genus) they identified usnic acid (UA) as a dibenzofurans representative, thirteen depsides, and eight depsidones [11]. Our previous studies proved that *U. barbata* ethanol extract also contains phenolic acids (caffeic acid, ellagic acid, p-coumaric acid, chlorogenic acid, cinnamic acid, and gallic acid) [12] and tannins [13].

Usnic acid (UA) is a well-known and abundant lichen secondary metabolite and can be isolated from *U. barbata* [14] and other *Usnea* sp. [10]. Various lichen bioactivities are due to this compound, such as gastroprotective [15], cardioprotective [16], neuroprotective [17], cytoprotective [18], cytotoxic [19], anticancer [20], antimicrobial [21], antidiabetic [22], analgesic-antipyretic [23], and anti-inflammatory [24] properties, mainly through its antioxidant action, reducing oxidative damage [25]. Some of the other phenolic metabolites previously mentioned also display different biological effects. Thus, from the depsides group, atranorin shows antimicrobial, anti-proliferative, cytoprotective, anti-inflammatory, and antioxidant activities [26]; lecanoric acid has antimicrobial [27], anti-proliferative [28], and antioxidant [29] effects; diffractaic acid shows analgesic-antipyretic [23], hepatoprotective [30], antimycobacterial [31], cytotoxic, and anti-proliferative action [32]. Depsidone’s representatives (norstictic, connorstictic, stictic, salazinic, and lobaric acid) have proven antioxidant, antimicrobial, and antitumor properties [33]. It has been reported that major *U. barbata* secondary metabolites have antioxidant potential [34]. Moreover, Rabelo et al. (2012) [35] proved that usnic acid displays a dual redox dose-dependent behavior; it protects the normal cells through an antioxidant effect and, at the same time, it induces tumor cell apoptosis by enhancing reactive oxygen species (ROS) levels.

Numerous studies have explored the bioactivities of lichen extracts in chemical solvents, such as acetone [36], ethanol [37], methanol [38], ethyl acetate [13], supercritical CO_2_ [39], chloroform [40], diethyl-ether, and petroleum ether [41]; only a few studies have described lichen extracts in natural solvents. Nowadays, the solvents with petroleum origin are restricted worldwide; therefore, the research on bio-based and renewable green solvents for extracting, purifying, and formulating natural products for the food and pharmaceutical industry has significantly increased [42]. According to the green chemistry concept, vegetable oils may become more competitive because of economic, food safety, and eco-friendliness concerns [43]. As solvents, vegetable oils are rich in bioactive compounds; there is a relationship between the solubilizing capacity of non-polar and polar bioactive components with the function of fatty acids and/or lipid classes and other minor components [41,42]. Yara-Varón et al. (2017) reported that various edible oils, such as soybean, olive, sunflower, corn, grapeseed, and canola oils, could extract carotenoids and antioxidants (astaxanthin), aromatic, and phenolic compounds [43]. From these common edible oils, one valuable vegetable oil containing various bioactive compounds is canola oil, the second most abundantly produced edible oil in the world, after soybean oil [44].

As cold-pressed rapeseed (*Brassica napus*) oil, canola oil (CNO) has a low erucic acid and glucosinolate content [45]. Triacylglycerols represent the major components (97–99%) in CNO, while other constituents (polyphenols, phytosterols, tocopherols, carotenoids, chlorophylls, phospholipids, monoglycerides, diglycerides, and free fatty acids) were quantified in a minor amount (1–3%) [46]. Canola oil shows a unique fatty acids profile: it is rich in monounsaturated fatty acids (MUFA, 68.6%) and polyunsaturated fatty acids (PUFA); CNO also significantly contains oleic acid (63.7%), linoleic acid (17.4%), and γ-linolenic acid (6.8%) [47]. Stearic acid, eicosenoic acid, erucic acid, arachidic acid, behenic, and palmitoleic acid are found in minor concentrations [47]. The saturated fatty acids content (SFA, 7.2%) in CNO is lower than in sunflower oil (11%) and soybean oil (15%) [48]. The bioactive constituents of canola oil are phenolic compounds, tocopherols, phytosterols, and carotenoids [47]. The phenolic compounds in CNO are mainly phenolic acids: sinapic acid, cinnamic acid, syringic acid, ferulic acid, 4-hydroxybenzoic acid, vanillic acid, and p-coumaric acid [49]. The heat treatment prior to rapeseed cold-pressing for the CNO obtaining process induces sinapic acid decarboxylation, resulting in canolol [50]; an oil-soluble bioactive phenolic compound that is easier to extract into CNO. It has significant antioxidant, antimutagenic, and anticarcinogenic effects [51]. Tocopherols of canola oil are γ-tocopherol and α-tocopherol, as the major tocopherols, while δ-tocopherol, plastochromanol-8 (PC-8), and β-tocopherol are the minor tocopherols [51]. The main CNO phytosterols are cholesterol, brassicasterol (specific for rapeseed oil), stigmasterol, campesterol, β-sitosterol, and Δ5 -avenasterol [51]. Finally, β-carotene, zeaxanthin, and lutein are the CNO carotenoids [46]. The hydroxyl group (–OH) of phenolic compounds, tocopherols, and phytosterols can scavenge the free radicals [47], with CNO being known for its considerable antioxidant properties.

In their previous study, Bassiouni et al. (2020) [52] prepared sunflower oil extract from *U. barbata* originated from Mexico; they determined UA content by ^1^H-NMR spectroscopy and compared its value with the one of a commercial *U. barbata* supercritical CO_2_ extract dissolved in sunflower oil. In addition, they comparatively investigated the antibacterial effects (on bacterial strains isolated from poultry) and the cytotoxic potential (on human tumor cell lines) of *U. barbata* sunflower oil extract and its previously obtained zinc salt precipitate.

Our research was performed on *U. barbata,* native to a peat bog zone in Romania’s highest volcanic mountains, the Călimani mountains [53]. The content of bioactive compounds represented the criterion in choosing the vegetable oil for the lichen extraction. Of the common edible oils, canola oil has a high content of phenolic compounds (especially phenolic acids) and significant antiradical activity (higher than soybean, corn, and sunflower oils). All studied properties were analyzed by comparing UBO with CNO to explore the impact of CNO-specific bioactive constituents’ interaction with lichen metabolites. This complex research, in fact, represents our study’s novelty. We aimed to investigate whether UBO could be a valuable bioactive extract due to a possible synergism between lichen metabolites and CNO phytoconstituents. Thus, using ICP-MS analysis for 23 metals, from thirteen elements quantified in dried *U. barbata* lichen (*d*UB), only six elements were detected in UBO. Five metals were common to both oil samples analyzed; chromium and nickel were over the permissible limits for edible oils and lower than those for plant medicines. The differences between UBO and CNO phytoconstituents were initially evidenced in an overlay of both FT-IR spectra; higher absorbances were observed in some peaks for UBO, suggesting lichen metabolites extracted by CNO. Then, using UHPLC’s considerable precision and power, an appreciable amount of usnic acid was quantified in *U. barbata* extract in canola oil. This developed UHPLC method was validated by evaluating the specificity, accuracy, precision, linearity, LOD, and LOQ and successfully used in UA determination. Both oil samples displayed a significant phenolic content, higher in UBO than in CNO, due to *U. barbata’s* phenolic metabolites. UBO also recorded the highest capacity for DPPH radical scavenging of all the autochthonous *U. barbata* dry extracts previously examined.

Moreover, we prepared emulsions from UBO and CNO for in vivo cytotoxicity evaluation on *A. salina* larvae. The obtained results proved that both oil emulsions have cytotoxic effects, inducing different progressive morphological changes, incompatible with larvae survival. After 6 h of brine shrimp nauplii exposure, UBO showed the highest cytotoxicity, significantly correlated with UA, phenolic content, heavy/trace metals levels, antiradical activity, and thixotropy (*r* > 0.984, *p* < 0.05). Nowadays, *U. barbata* is used as a homeopathic remedy, SBL *Usnea barbata* (CH 6, 30, 200, 1M, 10M) tincture (SBL Pvt. Ltd. Jaipur, Rajasthan, India); and as a dietary supplement, *Usnea* Liquid Extract, *Usnea barbata* Dried Thallus Tincture (Hawaii Pharm LLC, Honolulu HI, USA). Examining UBO and CNO rheological properties, our study aimed to investigate whether the extract in canola oil of *U. barbata* from the Călimani Mountains, Romania, could be an appropriate starting point for possible pharmaceutical formulations.

## 2. Results

### 2.1. Lichen Sample

Dried *U. barbata* lichen had a grey-green color, a fresh smell, and a spicy taste [13]. The obtained loss of drying value was 10.94 ± 0.94% for the dried lichen [13].

### 2.2. Mineral Analysis

The metal contents (µg/g) of the analyzed samples (UBO and CNO) are registered in Table 1.

The data from Table 1 show that five metals (Al, Ca, Cr, Mg, Ni) were quantified in CNO in different concentrations and lower than in UBO. Generally, the differences between the registered values are statistically significant (*p* < 0.05), excepting the calcium (Ca) and magnesium (Mg) content in both oil samples (*p* > 0.5). Thus, Ca and Mg contents had the most similar values (*p* > 0.5): 76.818 ± 14.289 µg/g and 6.951 ± 0.177 µg/g, respectively, in UBO; and 74.711 ± 4.048 µg/g and 6.852 ± 0.099 µg/g, respectively, in CNO. Aluminum (Al) showed a significant content in UBO (7.688 ± 0.086 µg/g), around 8-fold higher than CNO (0.975 ± 0.049 µg/g). In decreasing order, both trace metals nickel (Ni) and chromium (Cr) contents values were 0.713 ± 0.005 µg/g and 0.195 ± 0.005 µg/g, respectively, in UBO; and 0.339 ± 0.004 µg/g and 0.158 ± 0.002 µg/g, respectively, in CNO. Moreover, copper (Cu) was only found in UBO (0.155 ± 0.002 µg/g), being undetected in CNO (< 0.100 µg/g).

Our previous study quantified thirteen minerals in dried lichen (*d*UB) [21], of which only six elements (Table 1) were found in UBO, with considerably diminished content. Macro-elements were extracted in an insignificant amount in UBO. Therefore, from the 979.766 ± 12.285 µg/g Ca, 172.721 ± 0.647 µg/g Mg, and 101.425 ± 1.423 µg/g Mn quantified in *d*UB, in UBO were only found around 2.107 µg/g Ca and 0.99 µg/g Mg, with Mn being undetected. (Table 1). Aluminum decreased from 87.879 ± 1.152 µg/g in *d*UB, to 7.688 ± 0.086 in UBO; the Fe content of *d*UB was substantial (52.561 ± 2.582 µg/g); however, in UBO it was undetected. Three heavy/trace metals from *d*UB, Cu (1.523 ± 0.013 µg/g), Cr (1.002 ± 0.008 µg/g), and Ni (0.449 ± 0.011 µg/g), were also quantified in UBO: Cu and Cr with lower contents (0.155 ± 0.002 µg/g Cu and 0.047 µg/g Cr, respectively), with Ni in higher content (0.713 µg/g). It can be observed that the nickel content value in UBO is around the sum of *d*UB and CNO.

Only for Al, Cr, and Ni were the registered values (µg/g) in UBO significantly different than in CNO (*p* < 0.05). The differences in mineral content between UBO and CNO can be attributed to the metals extracted from the lichen; significant for aluminum, chromium, copper, and nickel (Table 1).

### 2.3. FT-IR Spectra

Fourier transform infrared (FT-IR) spectroscopy was performed to obtain the infrared spectra of both oil samples, and the absorption, emission, and photoconductivity, detecting specific functional groups in UBO compared to CNO. The overlay of FT-IR spectra of UBO and CNO, displayed in Figure 1, highlights the presence of *U. barbata* lichen compounds in UBO compared to CNO.

As can be seen, no appreciable differences between peaks of UBO and CNO were observed, apart from some distinctions in absorbances values. Both samples exhibited absorption bands at various wavenumbers, as follows: 725 cm^−1^, which could be assigned to the CH_2_ group vibration and the out-of-plane vibration of cis –HC=CH– group of disubstituted olefins [54]; 1099, 1168, 1245 cm^−1^, which could be related to the stretching vibration of the C–O ester groups [55]; 1379 cm^−1^, due to the bending symmetric vibration of C–H linkages of CH_2_ groups [56]; 1468 cm^−1^, from the bending vibration of C–H of CH_2_ and CH_3_ aliphatic groups [56]; 1654 cm^−1^, attributed to the C=C stretching vibration of cis-olefins [54,56]; about 1750 cm^−1^, due to the stretching vibration of the ester carbonyl functional groups of the triglycerides [54]; 2858 cm^−1^, probably from the symmetric stretching vibration of C–H of aliphatic CH_2_ group and 2930 cm^−1^, due to their asymmetric stretching vibration [55]; 2960 cm^−1^, attributed to the asymmetric stretching vibration of C–H of the aliphatic CH_3_ group [54]; 3011 cm^−1^, due to the C–H stretching symmetric vibration of the cis double bonds, =CH [56].

The peak observed at 2960 cm^−1^ is related to the C–H vibration of CH_2_ and CH_3_ groups, while the band at 1626 cm^−1^ led to the double C=C bonds found in the aromatic nucleus [20,57]. *U. barbata* extract in canola oil showed higher absorbances for the peaks at the 1468, 1654, 1700, 1750, 2858, 2930, 2960, and 3011 cm^−1^ wavenumbers.

### 2.4. UHPLC Determination of Usnic Acid Content in Usnea barbata Extract in Canola Oil

#### 2.4.1. Specificity

The usnic acid (UA) peak identity was confirmed by matching the analyte peak spectra and retention times extracted from chromatograms concerning the substance. Spectra analysis confirmed the identity of UA from UBO with the usnic acid standard. A peak purity index was obtained by spectral reprocessing using Chromera software, in a 240–700 nm range at a 15% peak height (Appendix A). The obtained value of the peak purity index was 1.48.

The obtained chromatograms of usnic acid standard in acetone, UBO in acetone, and two blanks (CNO in acetone and acetone) are represented in Figure 2 and Figure 3.

The retention time (RT) value for the usnic acid standard (50 μg/mL) at 282 nm was around 3.735 min, with the RT relative standard deviation (RSD%) value being about 0.115% (Figure 2a).

In UBO 10 mg/mL chromatograms (six injections), the usnic acid peak was revealed in RT range 3.731–3.736 min, proving the compound identity (Figure 3b and Appendix A). Other unknown peaks were recorded at RT 1.838–1.848 min, 2.444–2.452 min, and 2.573–2.585 min, similar to those of canola oil 10 mg/mL (Figure 2b and Figure 3a). Both blank solutions (canola oil and acetone) in the zone of the usnic acid peak, in the RT range of 3.7–3.9 min, did not show any signals (Figure 3a,b). Finally, in all chromatograms, the peak of acetone (the solvent for all standard solutions, control quality solutions (QC), blank, and samples) was reported at RT around 1 min ((Figure 3a,b) and Appendix A). Given all these data, our method could discriminate the studied compound (usnic acid) from other unknown constituents of the CNO matrix.

#### 2.4.2. Accuracy

Accuracy % was determined after six injections with QC1 (usnic acid 7.5 µg/mL in acetone) and calculated using the following formula (Equation (1)):(1)$$\mathrm{Accuracy}\;\%=\frac{\mathrm{CcQC}1}{\mathrm{CTQC}1}*100$$
where CcQC1 is the concentration of the injected QC1 solution; CTQC1 is the theoretical concentration of the QC1 solution. More details are presented in the Appendix A.

The data are shown in Table 2, and the obtained accuracy % value expressed as mean (*n* = 6 injections) ± SD was 96.614 ± 1.411%. This accuracy % value is included in the admissible limit range of 100 ± 10%.

The calculated spike recovery (%) value, as mean (*n* = 4 injections) ± SD, was 104.179 ± 1.373% (Table 3). The spike recovery (%) value is also included in the range of allowable limits, 100 ± 10% (Appendix A).

#### 2.4.3. Precision

Regarding the repeatability at the same concentration level, precision was measured using six injections of QC1 solutions (7.5 μg/mL UA in acetone) and another six injections with sample solution (UBO 10 mg/mL in acetone). The results are displayed as relative standard deviations (RSD%): UA RDS% = 1.631% and UBO RDS% = 3.104%. RDS% values meet the condition of acceptability, being lower than 5%. All these data are detailed in the Appendix A.

#### 2.4.4. Linearity

A five-point calibration curve (2.5; 5; 10; 25 and 50) was plotted in the range of 2.5 μg/mL to 50 μg/mL UA standard solutions from five repetitions for each UA concentration (Appendix A).

Furthermore, the coefficient of determination value (*R*^2^ = 0.995) proved the calibration curve’s admissibility condition (*R*^2^ > 0.99). The calibration curve is represented in Appendix A; the linear equation was the following (Equation (2)):(2)$$y=14.549\times 10^{3}\times x+\left(-12.621\times 10^{3}\right)$$

#### 2.4.5. Detection Limit (LOD) and Quantification Limit (LOQ)

The obtained data are registered in Table 4 and detailed in the Appendix A. This UHPLC method was validated for a LOD of 0.300 ug/mL and a LOQ of 1.250 ug/mL, given a S/N ratio of 6.00 and 19.18 μg/mL, respectively.

UHPLC determination was performed in triplicate, and the UA content (UAC) value is represented as the mean (*n* = 3) ± standard deviation (SD); UAC = 0.915 ± 0.018 mg/g UBO.

### 2.5. Total Phenolic Content

The absorbance values for both oil samples were read at 760 nm, with water as a blank (because the reagents were prepared using water as a solvent). The total phenolic content (TPC) values extracted in ethanol 96% were 2.592 ± 0.097 mg PyE/g in UBO and 2.243 ± 0.049 mg PyE/g in CNO. The TPC values obtained using acetone were 2.277 ± 0.057 mg PyE/g in UBO and 1.769 ± 0.039 mg PyE/g in CNO (Table 5).

### 2.6. Antioxidant Activity

The calculated % DPPH-radical scavenging value for UBO (82.182 ± 0.595%) was significantly different (*p* < 0.05) from that reported for CNO (64.806 ± 0.399%). Usnic acid 0.8 mg/mL in acetone had the lowest AA: 4.687 ± 0.025% DPPH-radical scavenging, significantly different compared to both oil samples (*p* < 0.05). The data displayed in Table 6 prove that UBO had a higher antioxidant activity than canola oil; the DPPH IC_50_ value for UBO is 0.942 ± 0.004 mg/mL and for CNO is 1.361 ± 0.008 mg/mL.

The correlation between TPC and AA (expressed as % DPPH-radical scavenger) was evaluated using linear trendlines, linear equations, and correlation coefficients (*R*^2^) for UBO and CNO (Table 6). The *R*^2^ values for UBO and CNO showed that their antioxidant activity is highly correlated with phenolic content (*R*^2^ > 0.900).

### 2.7. Cytotoxic Activity

Both emulsions examined under a microscope with different magnifications (40×, 100×, 400×) at 20 °C are displayed in Figure 4. It can be observed that the CNO emulsion is more uniform than that of UBO.

After 24 and 30 h, the viability of *A. salina* larvae exposed to negative controls (saline solution 0.3% and Poloxamer 407 5%) was 100%. The control solutions from the experimental pots had a regular aspect according to the second larval stage, swimming and showing the normally visible movements.

However, microscopic examination with magnitudes of 100× and 400× evidenced some slight changes in the digestive tract of the larvae exposed to Poloxamer 407 (Figure 5). These modifications consisted of an enlargement of the digestive tract and the presence of more lipid particles, with increased spaces between them (Figure 5g–i), compared to brine shrimp nauplii from the saline solution (Figure 5d–f).

After 24 h of exposure at the first dilutions (1:1, 2:1, 3:1) of UBO and CNO stock emulsions, all brine shrimp larvae were considered dead, because they did not show visible movements in experimental pots. By microscopical examination with 400× magnification, the coexistence of actually dead larvae and ones with fine antennae and peristaltic movements was only observed at UBO 1:1 and CNO 1:1 and 2:1 dilutions. Different and significant morphological changes, highlighted by microscopy at 100× and 400× magnifications and correlated with the dilution values, are shown in Figure 6. In Figure 6, it can be seen, first of all, that the larvae swallowed both UBO and CNO emulsions; second, it is evident that both emulsions caused a blockage in the digestive zone. The cytotoxic effect of CNO and UBO, leading to the death of *A. salina* larvae after 24 h, resulted from their penetration into the larvae’s body through the digestive tract. At 3:1 dilution, the tissue destruction phenomena were significantly increased in both samples.

The UBO-induced intestinal blockage weakened the digestive membranes, with epithelium structure disorganization in the terminal abdominal area, tissue destruction, expulsion of intestinal contents (at high concentrations), and finally, death of brine shrimp nauplii (Figure 6a–c,g–i). CNO caused a progressive digestion blockage, followed by a progressive invasion of the surrounding tissues with lipids. The *A. salina* larvae gradually died due to the increased body structure alterations induced by the aggressive emulsified lipid accumulation (Figure 6d–f,j–l). The lipid emulsification from both oil samples facilitates their passage through biological membranes and cell penetration. Thus, it is justified that the cytotoxicity phenomena were manifested relatively quickly, profoundly affecting the *A. salina* larvae in 24 h. Moreover, usnic acid and other cytotoxic lichen phenolic metabolites in UBO accelerated tissue damage, and larval death occurred faster than in the case of CNO. Summing the lipid accumulation with the canolol cytotoxic action also explains CNO’s intense effects on brine shrimp nauplii after 24 h.

All these observations led to the next step of the BSL assay, diminishing the larvae’s exposure time to only 6 h and simultaneously increasing the dilutions of both emulsions in the range 1:1–1:4. The results are displayed in Table 7 and Figure 7.

Data from Table 7 show the highest cytotoxicity (76.000 ± 5.354 %) for the UBO emulsion at 1:1 dilution (15% UBO). Thereby, the UBO cytotoxic action, in this case, is highly correlated (*r* > 0.984, *p* < 0.05) with UA, phenolic secondary metabolites, and heavy/trace minerals (Cr, Cu, Ni, Al) content. All these organic and mineral constituents are known for their cytotoxicity. Usnic acid, lichen phenolic metabolites, and copper are only found in UBO; Al, Ni, and Cr were quantified in UBO in higher content than in CNO. They could act synergistically with the CNO’s phytoconstituents and generate a significant cytotoxicity. Their penetration into the larvae cells was facilitated by the lipid emulsification of the *U. barbata* oil extract. The data from Table 7 show that UBO cytotoxicity did not vary proportionally with UBO concentration when the sample’s dilution increased. At 1:2 (10% UBO), larvae mortality significantly decreased to 44.583 ± 4.125%. Furthermore, higher UBO dilutions (1:3 and 1:4) induced similar mortality values as for CNO (24.912 ± 1.464 and 21.025 ± 1.450%, versus 21.801 ± 2.800 and 20.134 ± 1.652%).

The UBO cytotoxicity recorded approximately similar values, in the range 20–30%, with progressively diminished differences when the dilution was increased. Table 7 shows that 29.444 ± 3.425% was the highest mortality induced by CNO. The following values were fairly similar: 22.692 ± 2.059, 21.801 ± 2.800, and 20.134 ± 1.652 %, respectively.

Figure 7 shows the morphological changes induced by the UBO and CNO emulsions on *A. salina* larvae in 6 h of exposure. The microscopical images, showing that the differences in mechanism progressively decreased with the dilution increase (Figure 7i–p), also supporting the data from Table 7.

### 2.8. Color Evaluation

The color parameters of oil samples and the refractive indexes are displayed in Table 8. The UBO sample luminosity (*L**) was 46.597 ± 0.058 and shows a significant difference (*p* < 0.05) compared to CNO, which had a value of 49.293 ± 0.072. The green nuance described by the negative values of the *a** parameter was lower for UBO (−3.213 ± 0.006) than CNO (−3.950 ± 0.026). In contrast, the yellow nuance indicated by the positive values of *b** was more pronounced in CNO (29.040 ± 0.062) than in UBO (26.843 ± 0.038).

Hue angle (*h_ab_*) did not show significant differences (*p* > 0.05) among oil samples. Both samples were classified as greenish-yellow (>90°) and green color (<180°), according to the diagram with the sequence of colors according to hue angle [58]. On the other hand, Chroma (*C**) varied significantly, from 27.035 ± 0.038 for UBO, to 29.307 ± 0.065 for CNO. The color difference (Δ*E*) between UBO and CNO was 3.556 ± 0.095, which is considered a clear difference [49]. The refractive index did not highlight significant differences between the canola oil and *U. barbata* extract in canola oil.

### 2.9. Rheological Properties

The flow behavior of the oil samples was determined by steady shear measurement, as shown in Figure 8.

A proportional increase of shear stress (τ) with shear rate (γ) was observed in both oils. Their viscosity decreased to about 250 s^−1^ shear rate, showing a non-Newtonian behavior, and remained constant after these values, exhibiting a Newtonian-like liquid character from these points onward. The shear stress and viscosity values were similar for UBO and CNO, with no noticeable differences observed. Power-law model parameters also showed that no significant differences (*p* > 0.05) for consistency index (*K*) and flow index (*n*) between CNO and UBO were obtained (Table 7).

Thixotropy is a time-dependent shear thinning characteristic and is defined as ‘the continuous reduction of viscosity with time when shear stress is applied to a sample that has been previously at rest, and the subsequent recovery of viscosity in time when the shear stress is discontinued’ [59]. Both oil samples a exhibited time-dependent behavior, which was more pronounced in UBO than CNO (Figure 9a).

UBO presented a higher thixotropy area (32.763 ± 1.975 Pa·s) than CNO (17.430 ± 0.990 Pa·s), as shown in Table 7. 

The measurements conducted in oscillatory mode showed that the CNO and UBO samples presented higher loss modulus values (G″) than storage modulus (G′), which was expected; with a proportional increase of both moduli with frequency increase being observed (Figure 9b). The storage modulus measures the energy stored by the sample, while the loss moduli expresses the energy lost during deformation; a liquid material exhibiting values of G′ close to zero [60]. As expected, the G′ values of both oil samples were close to 0, with G″ values being much higher than G′, which indicated that the energy required to deform the sample was dissipated viscously. The rheological behavior was characteristic of a liquid [60]. The differences regarding G′ and G″ among samples were not appreciable, the values obtained at more than 5 rad/s being characteristic of a disorganized state [49].

The variations of apparent viscosity with temperature are shown in Figure 10.

As expected, both UBO and CNO viscosity values were strongly reduced with temperature increase, with the curve of apparent viscosity being higher for the region corresponding to the temperature increase of 20–70 °C. Minor differences among the oil samples’ viscosity behavior with temperature were observed.

### 2.10. Relationships between Characteristics

The similarities and differences among the considered characteristics are presented in Figure 11 and detailed in Appendix A.

The first principal component (PC1) explained 83.10% of the data variance, while the second one (PC2) explained 12.16% of the total variance, which quantified 95.26%. The total polyphenols contents, color parameters, usnic acid content, thixotropy, and some metal contents (Ni, Cu, Al, and Cr) were associated with PC1. In contrast, Ca, Mg, and the consistency index were associated with PC2. There was an opposition between *L**, *C**, *b**, and *h_ab_* color parameters and usnic acid content and TPC.

The most significant correlations (*r* > 0.878, *p* < 0.05) between the studied characteristics are presented in the Appendix A. Thus, a considerably positive correlation (*r* > 0.947, *p* < 0.05) was obtained between *a** parameter and DPPH radical scavenging activity, TPC, usnic acid, and Cr, Cu, Ni, and Al content. In contrast, all other color parameters were negatively correlated (*r* < −0.945, *p* < 0.05) with all the characteristics previously mentioned. The refractive index was positively correlated (*r* > 0.895, *p* < 0.05) with DPPH radical scavenging activity, TPC, usnic acid, Cr, Cu, Al, and Ni contents, and negatively correlated with all color parameters (*r* < −0.901, *p* < 0.05). The thixotropy was positively correlated (*r* > 0.878, *p* < 0.05) with DPPH radical scavenging activity, TPC, usnic acid, Cr, Cu, Ni, and Al contents. As expected, both TPC values were remarkably correlated (*r* > 0.948, *p* < 0.05) with DPPH radical scavenging activity and usnic acid content. Strongly positive correlations (*r* > 0.947, *p* < 0.05) were observed between TPC, usnic acid, Al, Cr, Cu, and Ni content. Finally, *A. salina* larvae mortality at 15% oil sample concentration showed the highest correlation (*r* > 0.984, *p* < 0.05) with TPC, usnic acid, Cr, Cu, Ni, Al contents, DPPH radical scavenging, and thixotropy.

## 3. Discussion

Based on green chemistry and green engineering principles, vegetable oils could become an ideal alternative solvent to extract various compounds [43]. They can be obtained from renewable resources at acceptable prices and be easily recycled. Goula et al. (2017) performed a green ultrasound-assisted extraction of carotenoids from pomegranate wastes using sunflower and soybean oils [61]; thereby, they obtained oils enriched with antioxidants. Li et al. (2014) [42] described a green oleo-extraction of natural volatile and non-volatile bioactive compounds from rosemary leaves with 12 refined vegetable oils (soybean, grapeseed, rapeseed, peanut, sunflower, olive, avocado, almond, apricot, corn, wheat germ, and hazelnut oils) and optimized them [62]. The *U. barbata* oil extraction was adapted from the simple and low-cost method presented by Bassiouni et al. [52]. They used 1000 mL sunflower oil; however, we opted for only 500 mL cold-pressed canola seed oil to approximately 20 g ground lichen.

Wickramasuriya et al. (2015) [63] showed that canola meal is a quality source of essential minerals; it contains Ca (0.68%), Mg (0.64%), Cu (10 µg/g), Fe (159 µg/g), Mn (54 µg/g), and Zn (71 µg/g). Of these six minerals, only Ca (74.711 ± 4.048 µg/g) and Mg (6.852 ± 0.099 µg/g) were quantified in canola oil; moreover, three other metals (Table 2) were found in low contents: Al (0.975 ± 0.049 µg/g), Cr (0.158 ± 0.002 µg/g), and Ni (0.339 ± 0.004 µg/g). These elements could have been accumulated due to the different technologies applied and vessels used in the canola oil obtaining process. According to international requirements for edible oils, the copper and nickel contents (0.713 ± 0.005 µg/g and 0.195 ± 0.005 µg/g, respectively) were higher than the permissible limits (0.1 µg/g Cu and 0.2 µg/g Ni [62]) in UBO; with only nickel (0.339 ± 0.004 µg/g) in CNO. Generally, when heavy metals are found in edible oils, the vegetable seed oil needs refining, to remove part of these for safe use [64]. Moreover, the admissible heavy metal levels of Cu and Ni in herbal medicines are 150  µg/g and 2.14  µg/g, respectively [65], higher than the Cu and Ni contents in UBO and CNO.

The FT-IR spectra evaluation showed that UBO exhibited higher absorbances in some bands. Thus, the appearance of a peak at 1626 cm^−1^ assigned to the C=C bonds from the aromatic nucleus and the increased absorbance in 2930 cm^−1^ wavenumber in UBO could be explained by the presence of usnic and placodiolic acids from *U. barbata* [11]. The absorption band at 1700 cm^−1^ in UBO, attributed to the carbonyl C=O group, also supports this hypothesis. The intense peaks observed in UBO at 2930, 2960, and 3011 cm^−1^ could also be related to the O–H stretching vibration of alcohols, polyphenols, carbohydrates, peroxides, and polysaccharides [66] extracted from the *U. barbata* dried lichen.

The usnic acid content of 0.915 mg/g UBO obtained using the UHPLC method corresponds to 2.162 mg% usnic acid in dried *U. barbata* lichen (Table 8). In another previous study, usnic acid was quantified in *U. barbata* acetone extract, and a similar UA concentration (2.115 mg %) was obtained [5]. Furthermore, Cansaran et al. (2006) [67] prepared six acetone extracts of *Usnea* sp of Anatolia (*U. florida*, *U. barbata*, *U. longissima*, *U. rigida*, *U. hirta*, and *U. subflorida*) by extracting 0.05 g of ground air-dried lichens with 10 mL acetone at room temperature. Using an HPLC method, they quantified usnic acid, obtaining 2.160 mg% UA content in *U. barbata* dried lichen (Table 8). In our research on the native lichen species *U. barbata* from the Călimani Mountains, Romania, we performed comparative studies on five dry extracts in different ‘preferable’ solvents, according to the green chemistry concept (acetone, ethyl acetate, ethanol, and methanol) [13]. They were obtained using Soxhlet extraction for 8 h, evaporating the solvent, and drying the extract in a niche for 12 h. In *U. barbata* dry extracts in ethanol, methanol, acetone, and ethyl acetate, the usnic acid content range was 127.21–376.73 mg/g (Table 8), notably higher than in UBO (0.915 mg/g). Finally, Zizovic et al. (2012) [68] and Ivanovic et al. (2013) [69] highlighted the advantage of high-pressure processing and supercritical fluid extraction (SFE) with carbon dioxide for usnic acid extraction from *U. barbata*. Using the same pressure (30 MPa) and different temperature values (40 °C and 60 °C, respectively), Zizovic et al. [68] obtained *U. barbata* extracts with minimal yields (0.60% and 0.38%, respectively) with significant usnic acid content (364.9 mg/g and 594.8 mg/g extract, respectively) in the Autoclave Engineers Screening System. The essential comparative data are displayed in Table 9.

Examining the data from Table 9, it can be seen that UBO shows 2.162 mg usnic acid/ 100 mg dried lichen, similar to UB-SFE obtained at 30 MPa, and 40 °C and 60 °C (2.190 and 2.226% UA), and higher than UB-SFE prepared at 50 MPa and 40 °C (1.243% UA). Moreover, *U. barbata* dry extract in ethyl acetate (obtained in our previous study) has a higher yield (6.27%), usnic acid content as mg/g extract (376.73 mg/g), and mg% in dried lichen (2.262%) than *U. barbata* obtained using SFE with CO_2_ at 40 °C (0.60% yield, UA = 364.9 mg/g UB-SFE and 2.190% UA) and a significantly lower cost price. Finally, UB-SFE obtained at 30 MPa and 60 °C recorded the highest UA content as mg/g extract (594.80 mg/g) but the lowest yield (0.38%).

Ivanovic et al. (2013) [69] performed various lichen thallus pre-treatments (in various mills: roller mill, ultra-centrifugal mill, and cutting mill ± rapid gas decompensation) and with different SFE conditions (pressure, temperature, CO_2_ pressure), aiming to obtain UB-SFE extracts with significant UA contents (Table 10).

Generally, it can be observed that the concentration of usnic acid in supercritical extract varies inversely in proportion with the extraction yield. The highest usnic acid content (648 mg/g UB-SFE) was obtained with ultracentrifugation and shearing as pre-treatments, and 40 °C, 50 MPa, and 992 m^3^/kg CO_2_ extraction conditions (Table 10). The UA content was significant, varying between 423–648 mm/g extract; the yield was very low (in the range 0.78–2.28%) and %UA/dried lichen of 0.479–1.243%.

The TPC values were 2.592 ± 0.097 mg PyE/g in UBO, and 2.243 ± 0.049 mg PyE/g in canola oil; two phenolic acids (cinnamic acid and p-coumaric acid) found in CNO were identified in *U. barbata* in another previous study [5]. In the sunflower extract of *U. barbata* (originated from Mexico), Basiouni et al. (2020) [29] recorded a phenolic content (4.4 mg ± 1.4 mg/mL) higher than the TPC determined in UBO. All the *U. barbata* dry extracts previously mentioned [13] displayed a total phenolic content in the range of 42.40–101.09 mg PyE/g, significantly higher than UBO (2.592 ± 0.097 mg PyE/g). However, their antioxidant activity was considerably lower (DPPH IC_50_ = 3.300–7.701 mg/mL) than that of UBO (DPPH IC_50_ = 0.942 ± 0.004 mg/mL). The remarkable antioxidant activity of UBO is due to the coexistence of the lichen phenolic secondary metabolites and canola oil bioactive constituents (phenolic acids, tocopherols, phytosterols, and carotenoids). The dried lichen extraction in canola oil could synergize cold-pressed rapeseed oil phytoconstituents with lichen secondary metabolites. Canola oil is rich in PUFA; in this case, the antioxidant activity could be defined as limiting or inhibiting nutrient oxidation (especially lipids and proteins) by restraining oxidative chain reactions [70]. Lichen phenolic compounds could delay the canola oil rancidity, avoiding lipid autooxidation; a fact also confirmed by the strong correlations obtained between scavenging DPPH activity, and TPC and usnic acid contents (*r* > 0.956, *p* < 0.05); they could break one of the phases of initiation or propagation of lipids autooxidation by hydrogen donation or electron transfer [71]. The dual redox behavior of usnic acid also justifies the high correlation (*r* > 0.984, *p* < 0.05) between antiradical activity and the cytotoxic effect of UBO.

Moreover, lichen phenolic metabolites in UBO and their interaction with CNO phytoconstituents significantly influenced the properties of the corresponding emulsions and the morphological changes in *A. salina* larvae, exposed to diluted samples for both 12 and 6 h. Simultaneously, the lipid emulsification from both oil samples shortened the time of passage through cell membranes and accelerated the penetration of usnic acid and other cytotoxic phenolic metabolites into the cells. This process explains how the cytotoxic phenomena with morphological changes and tissue damage, and ending with the death of *A. salina* larvae, were installed so quickly. Initially, based on the results of our previous studies [13], the sample dilutions were made referring to the usnic acid content estimated in each diluted sample. The exposure time was 24 h, as we utilized in previous studies on the other *U. barbata* extracts [13,72]. However, due to the mechanisms mentioned above, we did not quantify the cytotoxicity assay results gradually (dose-dependent) because the resulting mortality was 100% for all sample dilutions. Only by decreasing the exposure time from 24 to 6 h and the concentration of both extracts we managed to see different cytotoxic effects at dilutions with successively decreased usnic acid concentration. At the lowest dilution, 1:1, with the highest % UBO (15%), the death rate obtained showed the highest correlation (*r* > 0.984, *p* < 0.05) with oil extract mineral constituents, with four heavy/trace metals (Cr, Cu, Ni, and Al). This strong correlation could also have been the consequence of using the oil extract as an emulsion. With the accumulation of emulsified lipids, they penetrated and accumulated in the cells; thus, these metals synergistically act with lichen metabolites, inducing oxidative stress [73,74,75,76,77] and accelerating cell damage.

The color and rheological parameters of UBO and CNO are directly correlated to their constituents. Color parameters showed significant differences (*p* < 0.05) between UBO and CNO in luminosity, green nuance, yellow nuance, and chroma. The increase of darkness and green nuance, the decrease of color purity (described by chroma), and the diminishing of yellow nuance in UBO could be related to the colored constituents of *U. barbata* dried lichen (chlorophyll and phenolic compounds); a fact also supported by the significant correlations obtained in this study. The color parameters of various oil extracts in CIELab colorimetric systems have been previously studied [78]. The color difference (Δ*E*) between UBO and CNO was 3.556 ± 0.095; Buhalova et al. (2014) [79] obtained higher color differences when performing extraction of pine cones, oregano, and thyme with sunflower oil. They prepared oil extracts in a 1:5 ratio (herb: sunflower oil), keeping them in refrigerated conditions (0–4 °C) for six months. Δ*E* decreased in the following order compared to the control (sunflower oil): pinecone oil extract (21.6), thyme oil extract (13.0), and oregano oil extract (10.9). The refractive index did not register significant differences (*p* > 0.05) among both oil samples, because the specific gravity, molecular weight, and polarity of constituents in UBO did not change significantly compared to CNO; our results were close to those reported by Önal and Ergin for canola oil [80]. In the previously mentioned study, the pinecone extract had a decreased brightness and increased color difference compared to the control [79]; in our study, UBO showed diminished luminosity and recorded color differences compared to CNO as the control.

Rheological measurements are valuable analyses for oil extract flow behavior. Rheological property determination has an essential role in describing heat transfer, or in designing, evaluating, and modeling various treatments, with many applications in pharmaceutical [60] and food sciences [81]. Viscosity is the most important parameter evaluated by the rheological methods applied for various fluids, to identify their texture [82]. The rheological characteristics of oil extracts are influenced by many factors, such as temperature, shear rate, time, concentration, pressure, physicochemical properties, and phytoconstituents [83]; however, this type of flow is determined by temperature variation. Both UBO and CNO showed a proportional increase of shear stress with shear rate, and a decrease in viscosity with share rate increase up to 250 s^−1^, it being thought that this behavior could be due to the CNO long-chain molecules [60]. Flow index values closer to 1 indicate a viscous material, while values close to 0 are specific to elastic ones [84]; in our study, UBO and CNO displayed flow index values around 1.20, proving a viscous character. Aksoy et al. (2010) [85] reported a decrease of canola oil viscosity with temperature increase. Hojjatoleslamya et al. (2005) [86] studied the viscosity of some oils (soybean oil, sunflower oil, and canola oil) as a function of the shear rate; they also examined shear stress as a function of shear rate at different temperatures. The results obtained revealed that during heating, oils with a higher unsaturated fatty acid content (as soybean oil and canola oil) displayed a more rapid viscosity change with temperature than oils containing lower amounts of the previously mentioned acids (sunflower oil), due to their loosely packed structure [86]. Studying 12 vegetable oils (almond, canola, corn, grapeseed, hazelnut, olive, peanut, safflower, sesame, soybean, sunflower, and walnut), Fasina et al. (2006) [87] reported that their viscosities were positively correlated with MUFA content and negatively correlated with the amount of PUFA. Moreover, they stated that the viscosity of vegetable oils could be predicted, by knowing PUFA and/or MUFA contents. This aspect can be used in the pharmaceutical domain, in the selection and design of equipment and processes for using and storing vegetable oils and oil extracts.

A time-dependent viscosity change is the most desirable property in pharmaceutical formulations, due to their requirement of flexibility in drug delivery [88]. When the rheological manifestation of viscosity-producing structural changes is reversible and time-dependent, the effect is called thixotropy [89]. According to obtained data, the UBO thixotropy area (ΔA = 32.763 ± 1.975 Pa·s) being higher than CNO (17.430 ± 0.990 Pa·s) was due to the phenolic lichen metabolites quantified in UBO; their antioxidant activity could limit the incidence of irreversible structural changes in canola oil. These facts are also supported by the positive correlations between thixotropy and TPC and usnic acid content (*r* > 0.878, *p* < 0.05). Rheological pattern is also a valuable quality control procedure in pharmaceutical research, regarding drug formulation development for optimal delivery. Thus, this can justify the strong correlation (*r* > 0.984, *p* < 0.05) between cytotoxicity and thixotropy in oil extracts. While a 32 ± 0.5 °C value represents a commonly adopted parameter for the simulation of the stratum corneum conditions for topical pharmaceutical formulations, in the case of oral ones, a 37 ± 0.5 °C value is relevant for predicting the oils in vivo behavior [90]. The profile registered on the hysteresis curve is significant in the 32–37 °C range; UBO and CNO viscosity and viscoelastic behavior indicate a clear weakening of the intermolecular bond when the temperature increases, leading to a more liquid state, useful for an enhanced spreadability (in case of topical application) and flowability (in case of oral intake). Due to the change in activation energy, a linear relationship between viscosity and temperature can be seen. For oral pharmaceutical formulations, flowability is essential for ensuring a controlled deformation after intake; favorable for constituent release and increased compliance.

## 4. Materials and Methods

### 4.1. Materials

All chemicals, reagents, and standards used in our study were of analytical grade. Usnic acid standard 98.1% purity, DPPH, and Poloxamer 407 were purchased from Sigma-(Sigma-Aldrich Chemie GmbH., Taufkirchen, Germany); 65% HNO_3_, 30% H_2_O_2_, Folin-Ciocâlteu reagent, Pyrogallol, acetone, and ethanol were supplied by Merck (Merck KGaA, Darmstadt, Germany).

*Artemia* Brine Shrimp Eggs and *Artemia* salt (Dohse Aquaristik GmbH & Co. Gelsdorf, Germany) were bought online from https://www.aquaristikshop.com/ (accessed on 5 February 2022). Canola oil (TAF PRESOIL SRL, Cluj, Romania) was purchased from the manufacturer’s distribution units.

### 4.2. Lichen Extract Preparation

*U. barbata* thalli were harvested one by one from the branches of conifers in the Călimani Mountains (47°29′ N, 25°12′ E, and 900 m altitude) [53]—the highest Romanian volcanic mountains—in March 2020. The freshly collected lichen was separated from impurities; then, it was dried at 18–25 °C in a herbal room, protected from sunlight. Dried lichen preservation for an extended period was performed in similar conditions (Appendix A). The Department of Pharmaceutical Botany of the Faculty of Pharmacy, Ovidius University of Constanta, accomplished *U. barbata* identification using the standard methods [68]. It is preserved in the Herbarium of Pharmacognosy Department, Faculty of Pharmacy, Ovidius University of Constanta (*Popovici 2/2020, Ph-UOC*) [91].

To determine loss on drying for the lichen sample, a weighing ampoule was brought to a constant weight together with the lichen sample was kept in the oven at 105 °C for two hours; then cooled in a desiccator and weighed. The drying process continued in the oven for one hour [13], then cooled and weighed until a constant weight was achieved [92].

The extract of *U. barbata* in canola oil (UBO) was obtained using a method adapted from that described by Basiouni et al. (2020) [29]; from 20.2235 g dried and ground lichen (Appendix A) and 500 mL cold-pressed canola seed oil (TAF PRESOIL SRL, Cluj, Romania), in a dark place, at room temperature (21–22 °C). The brown container with both components was daily shaken manually for three months; after this period, UBO was filtered in a brown vessel with a sealed plug and preserved in a plant room, sheltered from sun rays. Both oil samples had a pH = 4.

### 4.3. Mineral Analysis

Both samples, CNO and UBO, were used for ICP-MS mineral analysis, according to European Pharmacopoeia 10.0 [70]; 23 metals were analyzed: Ca, Fe, Mg, Mn, Zn, Al, Ag, Ba, Co, Cr, Cu, Li, Ni, Tl, V, Mo, Pd, Pt, Sb, As, Pb, Cd, and Hg, using the ICP ability to generate charged ions from the metal species within the lichen sample [93]; thus, they were guided into a mass spectrometer and separated according to their mass-to-charge ratio (*m*/*z*). This ICP-MS method was detailed in our previous study [21].

The quadrupole inductively coupled plasma mass spectrometer was a NexION™ 300S (PerkinElmer, Inc., Hopkinton, MA, USA) with a triple cone interface and a four-stage vacuum system. This ICP-MS system is equipped with a universal cell with two gas lines (helium, ammonia, methane), which allows operation in collision mode (helium) and reaction mode (ammonia/methane) [21]. It is also equipped with a recirculating chiller (Perkin Elmer Shelton) and a peristaltic pumping system with acid-resistant tubing; 0.38-mm interior diameter (id) tubing for sample introduction and 1.3-mm id for drain exclusive. The samples were digested in mineralization Teflon vessels using Rotor 16HF100 in a PROSOLV microwave digestion system (Anton Paar GmbH, Graz, Austria), using a pressure-activated-venting concept. The Directed Multimode Cavity (DMC) enables highly efficient turbo heating with one magnetron in a compact system combined with a turbo cooling system, for rapid cooling from 180 °C to 70 °C [21]. The ICP-MS mineral analysis was performed using the kinetic energy discrimination (KED) method, measuring unit = counts per second (CPS). The peristaltic pumping system was washed with each sample (35 s), followed by a read delay (15 s) and the analytical phase. Finally, the peristaltic pump was washed with ultrapure deionized water (45 s). All processes involved an operation speed = 20–24 rotations / minute (rpm) [21].

The data obtained for mineral content were processed with Syngistix Software (PerkinElmer, Inc., Hopkinton, MA, USA) version 2.3 for ICP-MS. Mineral analysis was performed in triplicate, and the results are expressed as mean (*n* = 3) ± SD.

### 4.4. FT-IR Spectra Acquisition

FT-IR spectra of both oil samples were recorded in triplicate from 650 to 4000 cm^−1^ wavenumbers on a Thermo Scientific Nicolet iS20 (Waltham, MA, USA) spectrometer. The average spectra were used for interpretation. The resolution value was 4 cm^−1^. The data were processed with OMNIC software (9.9.549 version, Thermo Fisher Scientific, Waltham, MA, USA), and the characteristic bands were identified according to previous literature [52,53,62]. 

### 4.5. UHPLC Determination of Usnic Acid Content in Usnea barbata Extract in Canola Oil

In order to assess the concentration of usnic acid extracted in canola oil, the UHPLC method developed and validated in our previous studies [28] was adapted for this new purpose.

#### 4.5.1. Equipment and Chromatographic Conditions

In brief, the method for usnic acid quantification entailed analyzing the *U. barbata* extract in canola oil using a PerkinElmer^®^ Flexar^®^ FX-15 UHPLC system (PerkinElmer, Shelton, CT, USA) fitted with a Flexar FX PDA-Plus photodiode array detector (PDA). The UHPLC-PDA system is equipped with a Brownlee Analytical C18 column, having a length of 150 mm, an inner diameter of 4.6 mm, and filled with 5 µm porous particles (150 mm/4.6 mm, v5 µm). As a mobile phase, an isocratic methanol/water/glacial acetic acid (80:15:5) was used for 10 min per run, with a flow rate of 1.5 mL/min and an injection volume of 10 μL. The oven temperature was set to 25 °C, and the detection was made at 282 nm. All data analysis, peak purity, and processing were achieved using PerkinElmer Chromera^®^ CDS software (PerkinElmer, Inc, Hopkinton, MA, USA).

#### 4.5.2. Sample, Blank, Standard, and Quality Control (QC) Solutions Preparation

All requested solutions were prepared using acetone (Sigma-Aldrich Chemie GmbH, Taufkirchen, Germany) as a solvent; injection volume was lowered to 10 μL to avoid peak distortion.

The sample (UBO) and blank (CNO) solutions were made by diluting around 1 g UBO or CNO 100-fold in acetone (the concentration of obtained solutions was 10 μg/mL). Another blank solution was acetone; the solvent used for all solutions.

Five standard solutions were prepared using usnic acid at 98.1% purity (Sigma-Aldrich, St. Louis, MO, USA) by serial dilution in acetone: 50, 25, 10, 5, 2.5, 1.25, 0.612, 0.3 μg/mL.

Two quality control (QC) solutions, QC1 and QC2, were prepared. The first, QC1, consisted of usnic acid standard 7.5 μg/mL in acetone. QC2 was obtained by dissolving around 1 mg usnic acid in 1 mL CNO, followed by 100-fold dilution in acetone (the QC2 final concentration was 10 μg/mL, similarly to the sample and blank solutions).

#### 4.5.3. Validation of UHPLC Method

According to International Conference on Harmonization (ICH) guidelines (ICH Q2A 1994) [71], due to significant changes (sample matrix, sample preparation, injection volume, range of calibration curve), the UHPLC method parameters were revalidated for: specificity, precision, accuracy, linearity, the limit of detection (LOD), and limit of quantification (LOQ) [72].

#### 4.5.4. Specificity

Specificity was evaluated by injecting one acetone blank solution, one canola oil blank solution (the same oil that was used for sample extraction) diluted 100-fold in acetone to resemble the sample, one standard solution (50 μg/mL), and one unknown sample solution for which peak purity was determined in a 240–700 nm range at 15% peak height.

#### 4.5.5. Accuracy

Accuracy was estimated as the closeness of the experimental value to the actual amount by injecting six QC1 solutions of known concentration (7.5 µg/mL usnic acid in acetone). Moreover, the accuracy was expressed as spike recovery % by injecting four spike solutions prepared by dissolving usnic acid reference in canola oil (1 mg/mL) and diluting 100-fold (QC2). Results were expressed as the percentage accuracy by comparing the practical concentration to the theoretical one, considering the standard usnic acid purity of 98.1%. The accuracy, expressed as a percent of the spike recovery, was considered admissible when its value was 100 ± 10%.

#### 4.5.6. Precision

For the area repeatability at the same concentration level, precision was measured by injecting six control quality solutions 7.5 μg/mL (QC1) and 6 UBO solutions at 10 mg/mL concentration (100-fold dilution). This was expressed as relative standard deviation (RSD%), with an RDS% ≤ 5% acceptance criterion.

#### 4.5.7. Linearity

The linearity of the method was determined in the range 2.5–50 μg/mL, by calculating the coefficient of determination (*R*^2^) of the calibration curve (Appendix A) constructed from 5 repetitions for each point (Appendix A). An *R*^2^ value higher than 0.99 was considered an admissible criterion of linearity.

#### 4.5.8. Limit of Detection (LOD) and Limit of Quantification (LOQ)

LOD (S/N ≥ 3) and LOQ (S/N ≥ 10) were determined by injecting successively different UA standard solutions with low concentrations in decreasing order (1.250, 0.612, and 0.300 μg/mL) and calculated using the following formula (Equation (3)):(3)$$LOD=\frac{S}{N}=\frac{2H}{h}$$
where *S*—signal, *N*—noise, *H*—usnic acid peak height, and *h*—noise height in blank solution (acetone).

#### 4.5.9. Data Processing

Data analysis, peak purity determination, and processing were realized using PerkinElmer Chromera^®^ CDS software.

### 4.6. Total Phenolic Content

According to a previously described method, the total phenolic content was determined using Folin–Ciocâlteu reagent [13]. Pyrogallol was used as standard, the TPC values being calculated as µg of Pyrogallol equivalents (PyE) per g UBO (CNO). For this analysis, in two volumetric flasks of 25 mL, 5 mL of each CNO and UBO (A1 and A2) was added, completing up to the sign with ethanol 96%; B1 and B2 solutions were obtained. In two volumetric flasks of 25 mL, 2 mL of B1 and B2 solution was added. Then, 1 mL of Folin–Ciocâlteu reagent, 10 mL water, and 12 mL of 290 g/L of Na_2_CO_3_ solution were added up to the mark; in each volumetric flask, a blue coloration appeared. After 30 min of reaction in a dark place at room temperature, the absorbance values (each value being A1 in the calculation formula) were read at 760 nm, using a Jasco V630 UV-Vis Spectrophotometer (JASCO Corporation, Tokyo, Japan) with Spectra Manager™ software. A similar determination of phenolic contents was performed by dissolving UBO and CNO in acetone. The total polyphenols content (TPC) determination was performed in triplicate, and the obtained data were expressed as means (*n* = 3) ± SD.

### 4.7. Antioxidant Activity

The antioxidant activity of UBO and CNO was determined on a Jasco V630 UV-Vis Spectrophotometer (JASCO Corporation, Tokyo, Japan) using a DPPH free radical scavenging assay [28]. The DPPH solution was obtained by dissolution of DPPH (Sigma Aldrich, St. Louis, MO, USA) in methanol, to assess the absorbance value of 0.8 ± 0.02; then, 3.9 mL of DPPH solution with 0.1 mL of each oil sample were vortexed for 30 s. Their reaction time in a dark place at room temperature was 30 min; finally, the absorbances values at 515 nm were registered. The DPPH solution in methanol with no added extract was used as a standard, methanol as a blank, and canola oil and 0.800 mg/mL of usnic acid (0.02 g in 25 mL volumetric flask and up to the mark with acetone) as positive controls. Two dilutions in acetone were obtained (1:2 and 1:4) for UBO and CNO, and the scavenger activity was calculated according to Equation (4).
(4)$$\mathrm{Scavenging}\;\mathrm{of}\;\mathrm{DPPH}\;(\%)=100\times \frac{A\;control-A\;sample}{A\;control}$$

*A control* and *A sample* are the absorbencies values at 515 nm for DPPH and sample solutions. This determination was performed in triplicate; the obtained data are expressed as mean (*n* = 3) ± SD.

### 4.8. Cytotoxic Activity

#### 4.8.1. Preparation of CNO and UBO L/H Emulsions

Poloxamer 407 (Poly(ethylene glycol)-block-poly(propylene glycol)-block-poly(ethylene glycol)) is a non-ionic and water-soluble surfactant that is made up of a hydrophobic residue of polyoxypropylene (POP) between the two hydrophilic units of polyoxyethylene (POE) [94]. The FDA guide mentions Poloxamer 407 as an inactive ingredient for different pharmaceutical preparations (intravenous fluids, oral emulsions, suspension, ophthalmic, or topical formulations) [95]. It was selected for the present study due to its low toxicity and compatibility with cells, body fluids, and a wide range of chemicals [96]. 

Two L/H emulsions containing the same 30% *w*/*w* concentration of UBO and CNO, as oil phase fractions, were prepared. In the first step, the selected emulsifier agent, Poloxamer 407 (Sigma-Aldrich Chemie GmbH, Taufkirchen, Germany) at 5% *w*/*w* concentration, was dissolved into the amount of water required for obtaining 100 g of each emulsion. The dissolution was performed using a Heidolph MR 3001 K magnetic stirrer (Heidolph Instruments, Schwabach, Germany) at a low 300 rpm speed and room temperature. The oily phase was added slowly, the speed was increased to 1200 rpm, and the emulsion was stirred for 30 min until a homogenous binary system was achieved.

These emulsions were stored at 3–8 °C until use. Both emulsions had pH = 5.5.

#### 4.8.2. BSL Assay

Brine shrimp larvae were obtained by introducing the cysts of *A. salina* for 24–48 h, in a saline solution of 0,35%, under conditions of continuous light and aeration, at a temperature of 20 °C. After hatching brine shrimp in the first larval stage (instar I), they were separated and introduced into experimental pots (with a volume of 1 mL) in 0.3% saline solutions [13]. To obtain accurate results regarding the UBO cytotoxic effect, the analysis of the UBO emulsion was performed compared to the CNO one. *A. salina* larvae were not fed during the test, so as to not interfere with the tested extracts. This bioassay was valid for 24 h, during which the larvae had embryonic energy reserves as lipids.

Brine shrimp larvae were initially exposed to different UBO and CNO emulsions dilutions (emulsion: saline water) for 24 h (Table 11). As negative controls, the Poloxamer 407 aqueous solution 5% *w/w* concentration and saline aqueous solution 0.3% (pH > 7) were used.

After 24 h of exposure, the death rate (the larvae without visible movements in their experimental pots were considered dead) was the measurable parameter for quantifying larvae response to the various concentrations of CNO and UBO. These effects were also evaluated using microscopy with different magnifications and compared with negative controls.

The second step of the BSL assay consisted of lower dilutions preparation: 1:1, 1:2, 1:3, 1:4 from stock UBO and CNO emulsions in 0.3% saline aqueous solution (Appendix A). The brine shrimp larvae were exposed for only 6 h at these new dilutions; then, the cytotoxic effects were evaluated by counting the *A. salina* larvae without visible movements in their pots and examining with the microscope with different magnifications, for observing the presence or absence of fine movements (antennae and peristaltic movements) and morphological changes.

#### 4.8.3. Data Processing

The microscopic images were achieved using a VWR microscope VisiScope 300D (VWR International, Radnor, PA, USA) with Visicam X3 camera (VWR International Hilden, Germany) at 40×, 100×, and 400× magnifications and processed with VisiCam Image Analyzer 2.13. All observations were realized in triplicate.

### 4.9. Color and Refractive Index Evaluation

The color of oil samples was measured by reflectance using a Konica Minolta CR-400 (Konica Minolta, Tokyo, Japan) colorimeter. The luminosity (*L**), red (*a**+) or green nuance (*a**−), and yellow (*b**+) or blue nuance (*b**−) were recorded in triplicate. The hue angle (*h_ab_*), chroma (*C**), and difference of color (Δ*E*) against the control sample were calculated by using Equations (5), (6), and (7), respectively. Chroma measures the degree of difference against a grey color with similar lightness for each hue. In contrast, hue angle is the parameter that allows a color to be differentiated concerning a grey color with similar lightness [69].
(5)$$h_{ab}=arctan\left(\frac{b^{*}}{a^{*}}\right)$$
(6)$$C=\sqrt{{\left(a^{*}\right)}^{2}+{\left(b^{*}\right)}^{2}}$$
(7)$$\Delta E=\sqrt{\Delta L^{*2}+\Delta a^{*2}+\Delta b^{*2}}$$
where *h_ab_*—hue angle (0°—red, 90°—yellow, 180°—green, and 270°—blue), *C**—chroma (0—gray, 100—pure color), *L**—luminosity (0 for absolute black, 100 for absolute white), *a**—positive describes red and negative green nuance, *b**—positive represents yellow and negative blue nuance, Δ*E*—color difference against control, Δ*L** = *L^*^*_UBO_ − *L^*^*_Control_, Δ*a** = *a^*^*_UBO_ − *a^*^*_Control_, Δ*b** = *b^*^*_UBO_ − *b^*^*_Control_.

The refractive oil index was measured in triplicate using an Abbe portable refractometer (Kern Optics, Balingen, Germany) at 20 °C.

### 4.10. Rheological Properties

All rheological measurements were performed using a dynamic rheometer Thermo-HAAKE, MARS 40 (Karlsruhe, Germany) with a Peltier temperature controller, using cone (2°) and plate geometry of 35 mm diameter. The volume of the sample was 0.4 mL.

#### 4.10.1. Steady Shear Test

Measurements were done in the shear rate range of 0.1–1000 s^−1^ at a temperature of 20 °C, according to the method described by Yalcin et al. [30] with modifications. The oil sample was placed with a micropipette between the cone and plate, and the test was started immediately. The flow curve, shear stress versus shear rate, and viscosity in function of shear rate were obtained by raising the shear rate. The data for shear stress vs. shear rate were fit to the power-law model (Equation (8)):(8)$$\tau =K\cdot \gamma^{n}$$
where *τ*—shear stress (Pa), *γ*—shear rate (s^−1^), *K*—consistency index (Pa·s^n^), *n*—flow index.

#### 4.10.2. Thixotropy Loop

A thixotropic loop was obtained using a rotational rheological test; an increasing shear rate from 0 to 100 s^−1^ in 100 s was applied to the sample, then it was kept at 100 s^−1^ for 30 s, after that the shear rate was decreased from 100 to 0 s^−1^ in 100 s. The measurements were made at 20 °C. The areas under the upstream data points (A↑) and the downstream data points (A↓), and the thixotropy area (ΔA = A↑ − A↓) were calculated using Rheowin Data Manager software.

#### 4.10.3. Frequency Sweep Test

Before evaluating storage (G′) and loss moduli (G″) changes with frequency, the linear viscoelastic region (LVR) of each sample was determined in the 0.1–5.0 Hz frequency range. Then, a frequency sweep test was performed at 0.3 Pa (within the LVR) and 20 °C, in a frequency range of 0.628–6.283 rad/s, according to the protocol described by Yalcin et al. [30].

#### 4.10.4. Apparent Viscosity Variation with Temperature

For evaluating the influence of temperature on oil viscosity, the temperature was increased from 20 to 70 °C and decreased immediately to 20 °C, at a constant shear stress of 0.30 Pa and a frequency of 10 Hz. The heating rate was 5 °C/min, the method being adapted from that presented by Ennouri et al. [48].

#### 4.10.5. Data Processing

The data obtained for the rheological properties were processed using RheoWin software 4.80.0010 version (Thermo Haake, Karlsruhe, Germany).

### 4.11. Statistical Analysis

The differences among samples were evaluated using the Student *t*-test and ANOVA with Tukey test, with differences being considered significant at *p* < 0.05. Principal component analysis (PCA) with Pearson correlations were performed to identify the relationships between the studied characteristics. For this purpose, XLSTAT for Excel 2021 version (Addinsoft, New York, NY, USA) was used.

## 5. Conclusions

Our study results recommend the canola oil extract of *Usnea barbata* (L.) Weber ex F.H. Wigg from Călimani Mountains, Romania, as a valuable lichen extract, which could be a starting point for future pharmaceutical products. Further research could optimize the extraction process, to obtain a high metabolite content in *U. barbata* oil extract. Future studies could investigate other pharmacological activities due to synergic interaction between canola oil compounds and lichen secondary metabolites. Finally, the potential pharmaceutical applications of autochthonous *U. barbata* extract in canola oil could be explored, developing optimal formulations for delivery of its bioactive constituents.

## Figures and Tables

**Figure 1 plants-11-00854-f001:**
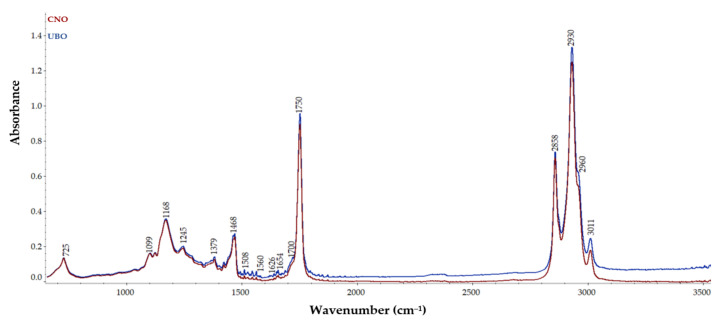
FT-IR spectra of *U. barbata* extract in canola oil (UBO) and canola oil (CNO).

**Figure 2 plants-11-00854-f002:**
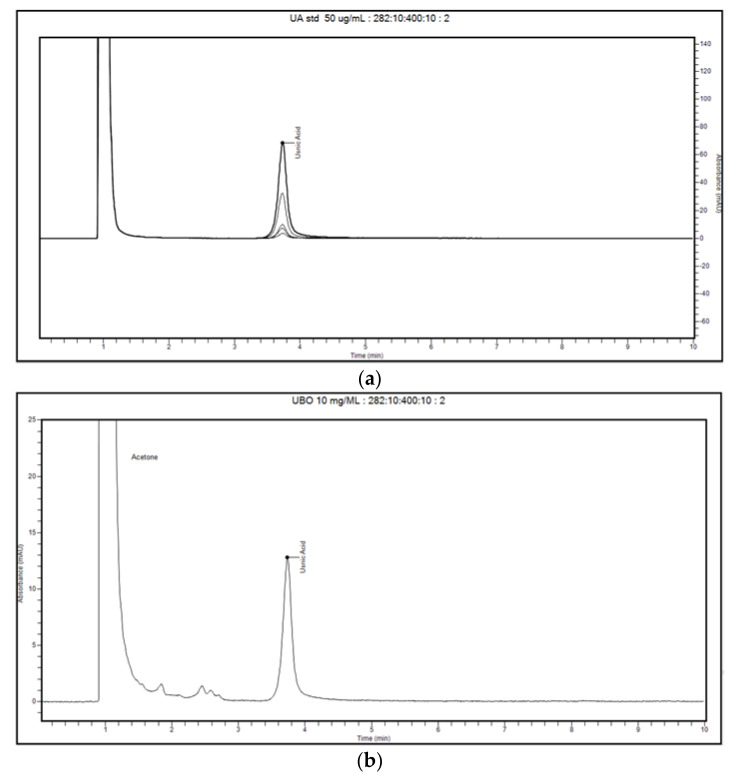
Chromatograms of usnic acid (UA) standards, an overlay of five peaks in a 2.5–50 μg/mL range (**a**), and *U. barbata* extract in canola oil (UBO) 10 mg/mL dissolved in acetone (**b**) at 282 nm.

**Figure 3 plants-11-00854-f003:**
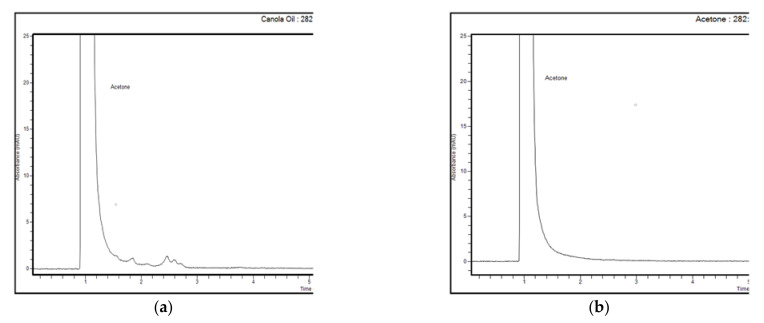
Chromatograms of blank solutions: canola oil 10 mg/mL in acetone (**a**) and acetone (**b**).

**Figure 4 plants-11-00854-f004:**
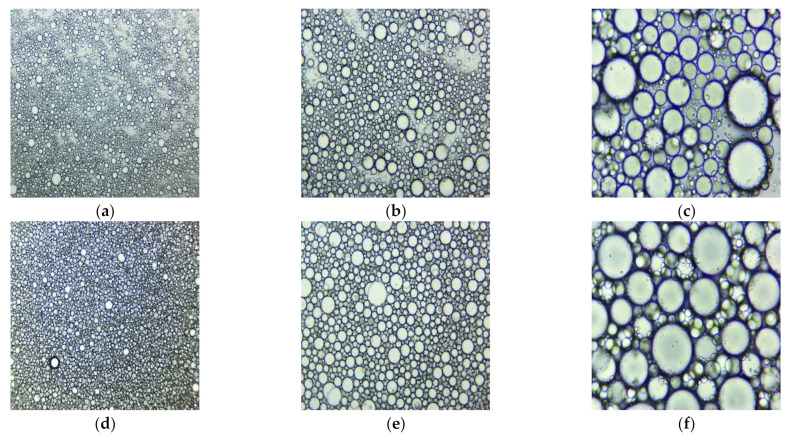
Microscopical images at 40×, 100×, and 400× magnification of UBO (**a**–**c**) and CNO (**d**–**f**), emulsions: 40× (**a**,**d**), 100× (**b**,**e**), and 400× (**c**,**f**) (20 °C).

**Figure 5 plants-11-00854-f005:**
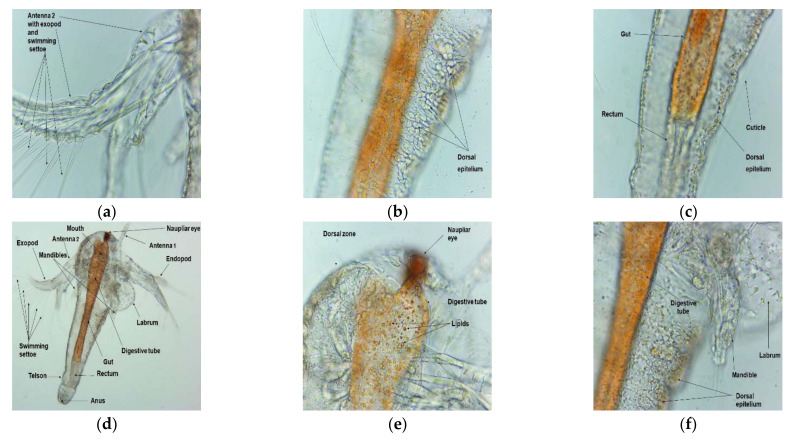
Microscopic images of *A. salina* nauplii morphology after 30 h (summing both steps times of BSL-assay, 24 h + 6 h) of exposure in saline solution 0.3% (**a**–**f**) and Poloxamer 407 at 3:1 dilution (**g**–**i**) with different magnifications: 100× (**d**,**g**) and 400× (**a**–**c**,**e**,**f**,**h**,**i**).

**Figure 6 plants-11-00854-f006:**
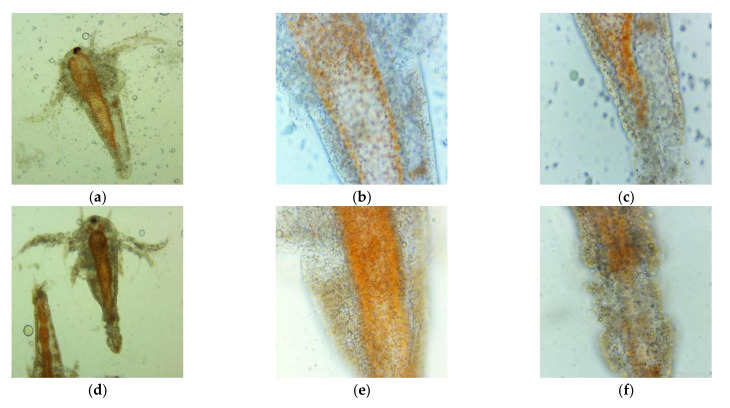
Morphological changes induced by UBO (**a**–**c**,**g**–**i**) and CNO (**d**–**f**,**j**–**l**): 1:1 (**a**–**f**) and 2:1 (**g**–**l**) on *A. salina* larvae after 24-h exposure, in microscopical images at different magnifications: 100× (**a**,**d**,**g**,**j**) and 400× (**b**,**c**,**e**,**f**,**h**,**i**,**k**,**l**).

**Figure 7 plants-11-00854-f007:**
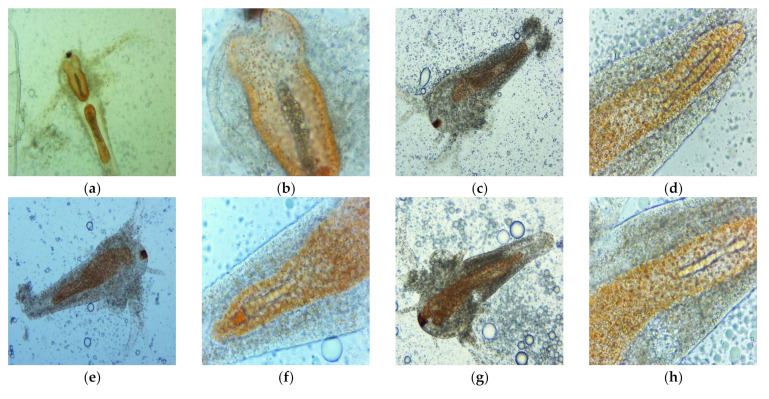
Morphological changes induced by UBO (**a**,**b**,**e**,**f**,**i**,**j**,**m**,**n**) and CNO (**c**,**d**,**g**,**h**,**k**,**l**,**o**,**p**) with different dilutions: 1:1 (**a**–**d**), 1:2 (**e**–**h**), 1:3 (**i**–**l**), 1:4 (**m**–**p**) on *A. salina* larvae after 6 h exposure, in microscopical images at different magnifications: 100× (**a**,**c**,**e**,**g**,**i**,**k**,**m**,**o**) and 400× (**b**,**d**,**f**,**h**,**j**,**l**,**n**,**p**).

**Figure 8 plants-11-00854-f008:**
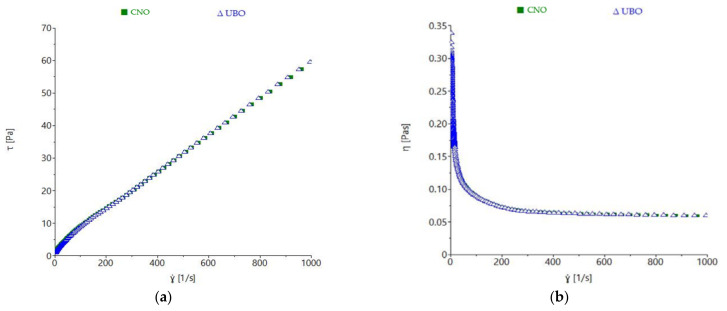
Influence of shear rate on (**a**) shear stress and (**b**) viscosity of UBO (*U. barbata* extract in canola oil) and CNO (canola oil).

**Figure 9 plants-11-00854-f009:**
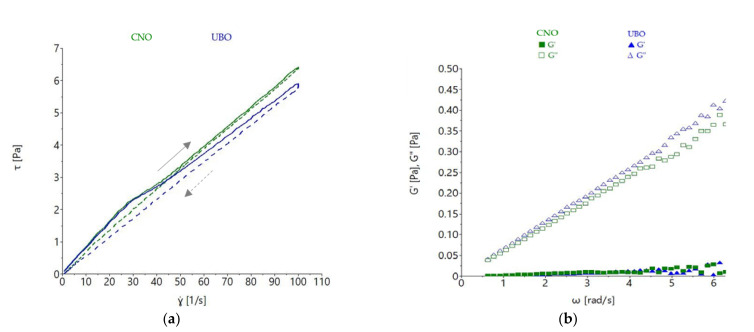
(**a**) Thixotropy loops of UBO—*U. barbata* extract in canola oil and CNO—canola oil, (**b**) variation of storage (G′) and loss (G″) moduli with frequency: UBO—*U. barbata* extract in canola oil, CNO—canola oil.

**Figure 10 plants-11-00854-f010:**
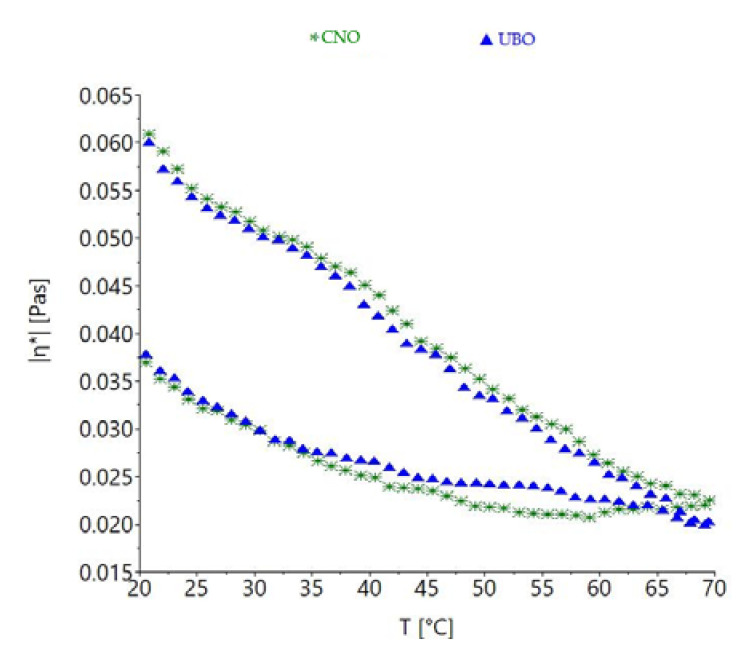
Variation of apparent viscosity (η*) with temperature: UBO—*U. barbata* extract in canola oil; CNO—canola oil.

**Figure 11 plants-11-00854-f011:**
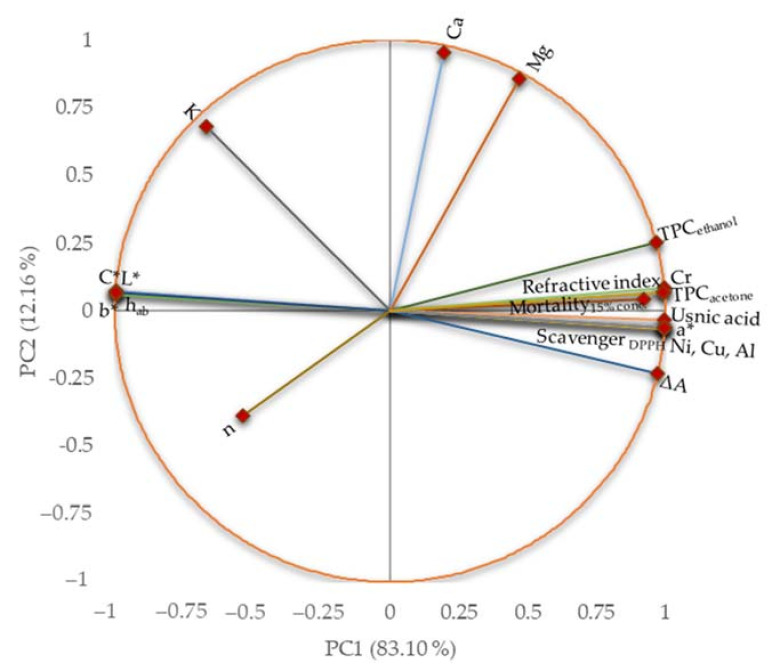
Principal component analysis (PCA) plot for UBO and CNO characteristics: *L**—luminosity, *a** and *b**—color indicator parameters, *h_ab_*—hue angle, *C**—chroma (color intensity indicator), K—consistency coefficient, *n*—flow index, ΔA—thixotropy area, Al—aluminum, Ca—calcium, Cr—chromium, Cu—copper, Mg—magnesium, Ni—nickel, Mortality 15% *conc*.—mortality of *A. salina* larvae at 15% oil sample concentration, TPC—total phenolic compounds.

**Table 1 plants-11-00854-t001:** The metals contents in dried *U. barbata* lichen (*d*UB), *U. barbata* extract in canola oil (UBO) and canola oil (CNO).

Metal	*d*UB	UBO	CNO	LOQ
Al (µg/g)	87.879 ± 1.152 ^a^	7.688 ± 0.086 ^b^	0.975 ± 0.049 ^c^	1.000
Ca (µg/g)	979.766 ± 12.285 ^a^	76.818 ± 14.289 ^b^	74.711 ± 4.048 ^b^	5.000
Cr (µg/g)	1.002 ± 0.008 ^a^	0.195 ± 0.005 ^b^	0.158 ± 0.002 ^c^	0.100
Cu (µg/g)	1.523 ± 0.013 ^a^	0.155 ± 0.002 ^b^	ND	0.100
Mg (µg/g)	172.721 ± 0.647 ^a^	6.951 ± 0.177 ^b^	6.852 ± 0.099 ^b^	5.000
Ni (µg/g)	0.449 ± 0.011 ^b^	0.713 ± 0.005 ^a^	0.339 ± 0.004 ^c^	0.100

*d*UB—dried *U. barbata* lichen, UBO—*U. barbata* extract in canola oil, CNO—canola oil, LOQ—limit of quantification, Al—aluminum, Ca—calcium, Cr—chromium, Cu—copper, Mg—magnesium, Ni—nickel, ND—non-detected (when metal content < LOQ). Mean values followed by different superscript letters (^a^–^c^) are significantly different (*p* < 0.05).

**Table 2 plants-11-00854-t002:** Accuracy % calculation after six injections with 7.5 µg/mL UA QC1 solution.

Injection Number	CcQC1 (µg/mL)	CTQC1 (µg/mL)	Accuracy (%)
1	6.946	7.350	94.501
2	7.241	7.350	98.521
3	7.132	7.350	97.039
4	6.987	7.350	95.060
5	7.119	7.350	96.859
6	7.181	7.350	97.705
Average (*n* = 6)	7.101	7.350	96.614
SD	0.104	-	1.411

SD—standard deviation, QC1—control quality solution 1, Cc —concentration of the QC1 injected, CT—theoretical concentration of QC1 (7.5 µg/mL usnic acid in acetone).

**Table 3 plants-11-00854-t003:** Spike recovery % calculation, after four injections with 10 µg/mL usnic acid QC2 solution (spike solution).

Spike Solution Number	CcQC2 (µg/mL)	CTQC2 (µg/mL)	Spike Recovery (%)
1	10.388	9.810	105.810
2	10.134	9.810	103.262
3	10.059	9.810	102.446
4	10.328	9.810	105.199
Average (*n* = 4)	10.227	9.810	104.179
SD	0.135	-	1.373

SD—standard deviation, QC2—control quality solution 2 (spike solution), Cc—concentration of the spike solution (QC2) injected, CT—theoretical concentration of QC2 (calculated considering usnic acid standard purity = 98.1%). QC2 was obtained by dissolving around 1 mg usnic acid in 1 mL CNO, followed by 100-fold dilution in acetone (the QC2 final concentration was 10 μg/mL, similar to the sample and blank solutions.

**Table 4 plants-11-00854-t004:** Calculation of detection limit (LOD) and quantification limit (LOQ) values (μg/mL).

Standard Solution Concentration (UA, μg/mL)	Noise N = h	H	Signal (S = 2H)	LOQ, LOD (μg/mL)
1.250	0.082	0.783	1.566	19.187
0.612	0.082	0.308	0.616	7.544
0.300	0.082	0.245	0.490	6.001

LOD (S/N ≥ 3) and LOQ (S/N ≥ 10) were determined by injecting successively different UA standard solutions with low concentrations in decreasing order (1.250, 0.612, and 0.300 μg/mL); S—signal, N—noise, H—usnic acid peak height, h—noise height in blank solution (acetone).

**Table 5 plants-11-00854-t005:** Total phenolic content in UBO and CNO.

Sample	UBO	CNO
*Solvent*	*TPC (mgPyE/g UBO/CNO)*	
Ethanol 96%	2.592 ± 0.097 ^a^	2.243 ± 0.049 ^b^
Acetone	2.277 ± 0.057 ^a^	1.769 ± 0.039 ^b^

UBO = *U. barbata* extract in canola oil, CNO = canola oil, mg PyE/g = mg equivalents pyrogallol/g. Mean values followed by different superscript letters (^a^ and ^b^) are significantly different (*p* < 0.05).

**Table 6 plants-11-00854-t006:** Antioxidant activity of UBO and CNO, and correlation between TPC and AA in both oil samples.

Parameter	UBO	CNO
DPPH IC50 (mg/mL)	0.942 ± 0.004 ^a^	1.361 ± 0.008 ^b^
% DPPH-radical scavenging	82.182 ± 0.595 ^a^	64.806 ± 0.399 ^b^
Linear equation	y = 23x + 28.336	y = 24.654x + 16.448
*R* ^2^	0.996	0.919

UBO—*U. barbata* extract in canola oil, CNO—canola oil control sample, TPC—total phenolic content, µg PyE/g—µg equivalents pyrogallol/g UBO, AA—antioxidant activity, *R*^2^—correlation coefficient. The means values followed by different superscript letters (^a^ and ^b^) are significantly different (*p* < 0.05).

**Table 7 plants-11-00854-t007:** The results obtained from the BSL assay after 6 h of *A. salina* larvae exposure, at different dilutions of UBO and CNO emulsions.

Oil Sample		% Mortality *	
*Dilution*	*% Oil sample* *Concentration*	*UBO*	*CNO*
1:1	15	76.000 ± 5.354 ^ax^	29.444 ± 3.425 ^bx^
1:2	10	44.583 ± 4.125 ^ay^	22.692 ± 2.059 ^bxy^
1:3	7.5	24.912 ± 1.464 ^az^	21.801 ± 2.800 ^axy^
1:4	6	21.025 ± 1.450 ^az^	20.134 ± 1.652 ^ay^

* As a selection criterion, *A. salina* larvae were considered dead when their movements were not visible in experimental pots; UBO—*U. barbata* extract in canola oil, CNO—canola oil. The 1:1, 1:2, 1:3, and 1:4 represent the ratio between UBO/CNO emulsions and 0.3% saline aqueous solution, measured as volume units (µL). Mean values followed by different superscript letters (^a^ and ^b^ in the same row for samples comparison and ^x^–^z^ in the same column for dilutions comparison) are significantly different (*p* < 0.05).

**Table 8 plants-11-00854-t008:** Color evaluation and rheological properties of UBO and CNO.

Characteristic	UBO	CNO
*Physical properties*		
*L** (adim.)	46.597 ± 0.058 ^b^	49.293 ± 0.072 ^a^
*a** (adim.)	−3.213 ± 0.006 ^a^	−3.950 ± 0.026 ^b^
*b** (adim.)	26.843 ± 0.038 ^b^	29.040 ± 0.062 ^a^
*h_ab_* (°)	178.548 ± 0.000 ^a^	178.564 ± 0.001 ^a^
*C** (adim.)	27.035 ± 0.038 ^b^	29.307 ± 0.065 ^a^
Δ*E* (adim.)	3.556 ± 0.095	-
Refractive index (adim.)	1.4715 ± 0.000 ^a^	1.4710 ± 0.000 ^a^
*Power-law model parameters describing oil flow behavior*		
K (Pa·sn)	2.153 ± 0.006 ^a^	2.160 ± 0.000 ^a^
*n* (adim.)	1.203 ± 0.015 ^a^	1.220 ± 0.020 ^a^
*R* ^2^	0.999	0.999
*Thixotropy*		
ΔA (Pa·s)	32.763 ± 1.975 ^a^	17.430 ± 0.990 ^b^

UBO—*U. barbata* extract in canola oil, CNO—canola oil, adim.—adimensional, *L*—luminosity, *a** and *b**—color indicator parameters, *h_ab_*—hue angle, *C**—chroma, color intensity indicator, Δ*E*—color difference, K—consistency coefficient, *n*—flow index, ΔA—thixotropy area. Mean values followed by different superscript letters (^a^ and ^b^) are significantly different (*p* < 0.05).

**Table 9 plants-11-00854-t009:** Various extraction processes with different conditions correlated with usnic acid content expressed as mg/g extract and % UA in dried *U. barbata* lichen.

*U. barbata* Extract	Extraction Solvent	Pressure (MPa)	Temperature of Extraction (°C)	Yield %	UAC (mg/g *U. barbata* Extract)	% UA in Dried Lichen
UB-SFE ^a^	99% CO_2_	30	60	0.38	594.80	2.226
UBDEA ^b^	Ethyl acetate		75–80	6,27	376.73	2.362
UB-SFE ^a^	99% CO_2_	30	40	0.60	364.90	2.190
UBDA ^b^	Acetone		55–60	6.36	282.78	1.798
UBDM ^b^	Methanol		65	11.29	137.60	1.553
UBDE ^b^	96% Ethanol		75–80	12.52	127.21	1.592
UBO	Canola oil		22		0.915	2.162

UB-SFE—*U. barbata* extract obtained by supercritical fluid extraction with CO_2_, UBDEA—*U. barbata* dry extract in ethyl acetate, UBDA—*U. barbata* dry extract in acetone, UBDE—*U. barbata* dry extract in ethanol, UBDM—*U. barbata* dry extract in methanol, UBO—*U. barbata* extract in canola oil, UBA—*U. barbata* extract in acetone, ^a^ [68], ^b^ [13].

**Table 10 plants-11-00854-t010:** The influence of lichen thalli pre-treatment and extraction conditions on usnic acid (UA) content and extraction yield in UB-SFE [69].

Pression (MPa)	Temperature (°C)	CO_2_ Pressure (m^3^/kg)	Pre-Treatment	Yield %	UAC (mg/g Extract)	%UA in Dried Lichen
30	40	911	RM	0.78	634.5	0.481
			UM	1.27	617	0.806
			CM	0.86	558.1	0.479
			UM + RGD	1.46	423	0.618
50	40	992	UM	0.85	648	0.551
			UM + RGD	1.50	645	0.968
			RM	1.67	585	0.977
			CM	2.28	545	1.243

UB-SFE—*U. barbata* extract obtained by supercritical fluid extraction with CO_2_, RM—roller mill; UM—ultra-centrifugal mill; CM—cutting mill; RGD—rapid gas decompression.

**Table 11 plants-11-00854-t011:** The samples and control dilutions and/or concentrations used in BSL assay for 24 h.

Sample	UBO Emulsion			CNO Emulsion		Poloxamer 407.5%	
*No*	*Dilution*	*%UBO*	*UA mg/mL*	*Dilution*	*%CNO*	*Dilution*	*%Poloxamer 5%*
1	1:1	15	0.122	1:1	15	1:1	35
2	2:1	20	0.162	2:1	20	2:1	46.666
3	3:1	22.5	0.183	3:1	22.5	3:1	52.5

UBO—*U. barbata* extract in canola oil; CNO—canola oil; UA—usnic acid; the 1:1, 2:1, 3:1 represent the ratio between UBO emulsion/CNO emulsion/Poloxamer 407 and 0.3% saline aqueous solution, measured as volume units (µL).

## Data Availability

Not available.

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
