# Peer review of "Antioxidant, Cytotoxic, and Rheological Properties of Canola Oil Extract of Usnea barbata (L.) Weber ex F.H. Wigg from Călimani Mountains, Romania"

_plants, 2022, doi:10.3390/plants11070854_

Round 1

Reviewer 1 Report

Thank you for the revision. The efforts of the authors are greatly appreciated. The manuscript is in a better shape now.

However, my previous concern about the suitability of this paper being accepted still persists. Personally, I think this work will be better received by readers of journals dealing with food science or related disciplines. But I would like to leave this issue for the editor to decide.

Reviewer 2 Report

This revised manuscript has been modified according to the reviewer’s comments. It is acceptable for publication.

Reviewer 3 Report

The Authors have improved their manuscript considerably in terms of presentation of the results and the analytical techniques used. They applied complex method combination to characterise the plant extract. However, as mentioned in the previous review, the weak point of the research is sampling, i.e. the number of plants collected. As this is not specified in the present manuscript, the reviewer can only suppose that this did not change since the previously submitted paper. Unfortunately, results deduced from the analysis of a few number of plants taken from a restricted geographical location can not be generalized in terms of either antioxidant properties, elemental composition or cytotoxicity. Eventually, the research could explore more the the synergistic effects of the plant and canola oil combination and the technological aspects of the extraction, without generalizing the results obtained from the analysis of a small-sized plant sample.

Round 2

Reviewer 3 Report

Dear Authors, 

Thank you for your consistent improvements performed on the manuscript. As a final suggestion, please, delimitate the harvesting zone even better by indicating its GPS coordinates.

Author Response

This manuscript is a resubmission of an earlier submission. The following is a list of the peer review reports and author responses from that submission.

Round 1

Reviewer 1 Report

Investigated material is highly pharmacologically important and the idea of extracting U. barbata using canola oil is interesting. However, the study lacks innovation. My recommendations are as follows:

Even though in the Introduction section there are a lot of data included, I recommend including more specifically and clearly what is innovative in this study and what is the contribution. For example, authors cited a study where lichen Usnea barbata extract was obtained using sunflower oil and the same extraction procedure. What should be emphasized is the innovate potential of their study compared to the referred study.  

The biggest disadvantage of this study is the procedure for preparation of the extract. I recommend trying to explain and justify why the authors chose this extraction approach.

Extraction (at room temperature) with extraction time 3 months and daily shaken (probably manually) is an overcome method since more efficient extraction techniques were developed. Additionally, the extraction is conducted just once, without replications. Room temperature for three months is not a precisely defined extraction parameter since there can be serious variations in temperature.

The procedure is not so optimal for application. Potentially, by increasing mixing, extraction efficiency can be increased with reducing extraction time. Also, an interesting approach can be investigation of different temperature and extraction time on properties of extracts and determination of optimal temperature and time which can provide the most optimal properties and avoid degradation. Therefore, it would be highly useful to apply same analyses with the goal to provide the most optimal extraction parameters and the highest quality of extracts.

Additionally, particle size is a very important parameter, so, if the material was grounded, please include particle size and information regarding equipment for grounding. Also, please include moisture content of the material. 

It was stated, “bioactivities of lichen extracts in chemical solvents as acetone [25], ethanol [26], methanol [27], ethyl acetate [28], and in only a few studies, oil extracts are described.” However, there is also supercritical carbon dioxide extraction, which is superior for extraction of lipophilic compounds. Therefore, my recommendation is to include more studies where innovative and alternative techniques are applied.

Include explanation for DPPH.

Row 753, 773 I recommend consistent use of abbreviation U. barbata (after the initial use of the abbreviation)

Row 455-464 Here the content of usnic acid is compared with different studies. However, I recommend including at least general extraction parameters for cited studies (time, temperature, methods, etc.).

Some of the metals are with higher contents than permissible limits. Does that mean that this extract cannot be used? How can this problem be overcome?

Reviewer 2 Report

This manuscript describes the physical characters (e.g. color, refractive index, viscosity) of a Canola oil extract of the title lichen material and its antioxidant potential. Overall, the methods used are standard and appropriate; and the results are reliable. There is no concern about the experimental findings. However, this reviewer has the following questions:

1) I am not sure the contents of this manuscript fall into the scope of the Polyphenol special issue. Conceivably the oil extract would contain a lot of different classes of constituents other than polyphenols. There is no evidence to demonstrate correlations between the polyphenol contents and the physical characters nor the antioxidant activity of the oil preparation, although it is well known that polyphenols are antioxidants. 

2) The so-called Total Polyphenol Content assay is actually a measurement of total phenols, not limited only to polyphenols.

3) This reviewer opines that readers of the Plants would have limited interest in this manuscript. It might be suitable for publication in journals specialized in food science or nutrition.

Reviewer 3 Report

This manuscript is not recommended for publication. This is because the content of this manuscript lacks novelty and is fragmented. Originally, the content of this manuscript should be included in the authors' recent report (Plants 2021, 10, 909. https://doi.org/10.3390/plants10050909). Below is a list of comments on this manuscript.

  1. The "Abstract" in this manuscript is redundant. The authors should describe the essence of this study concisely and clearly.
  2. The "Introduction" in this manuscript is redundant. The authors should clearly describe the purpose and background of this study, similar examples of studies in the past, and the novelty of this study.
  3. The authors should clearly explain why canola oil was used as the extraction solvent for the components of Usnea barbata. In the report by Basiouni et al. (Metabolites 2020, 10, 353; doi:10.3390/metabo10090353) cited in this manuscript, sunflower oil is used as the extraction solvent for the components of Usnea barbata. The authors should clarify the novelty of this study by mentioning the difference in the composition of these oils used for extraction and the relationship between the extracted components.
  4. The authors use the scavenging activity of DPPH radicals to evaluate antioxidant activity. Although this method is simple, the radical scavenging activity is only evaluated from the decrease in the absorbance. The authors should investigate using other antioxidant activity evaluation systems.
  5. The authors report in this manuscript a comparison of the physical properties of canola oil and its extracts. The authors should explain the novelty of these results.

Reviewer 4 Report

The manuscript Characterization of Usnea barbata (L.) Weber ex F.H. Wigg Extract in Canola Oil fits the journal’s scope. The authors present the obtaining of an Usnea barbata macerate in rapeseed oil. The design of the research is simple and presented in sufficient detail. The methods could be presented more briefly, and some sections of the results (especially the UHPLC Determination of Usnic Acid Content validation) could be included in the supplementary material. The aim of the study and the novelty of this research are not clearly stated. Thus, the discussions and the conclusions should be amended accordingly.

Other major concerns:

The conclusions should be re-written.

Please justify the rationale of using FT-IR Spectra analysis

Minor :

Lines 141-143 – please indicate the reference

The abstract should be shortened.

440-454 – the paragraph should be deleted

Please indicate the voucher specimen number

The methods could be presented more briefly.

Please indicate the number of determinations of FT-IR Spectra Acquisition

Please re-arrange the material and methods sections. The materials should be presented in a separate subsection at the beginning of the section (for example lines 661-662).

Reviewer 5 Report

The Authors performed a complex research helping a better understanding of the effect of a particular lichen of potential pharmacological interest.

Some remarks and questions:

  • L123-124: please, refrain from ambiguous claims like "CNO is considered one of the healthiest vegetable oils"
  • Canola oil is reported as containing significant amounts of omega-3 acid, i.e. alpha-linolenic acid. Please, revise the part referring to fatty acid composition.
  • The number of digits is inadequate and not justified for spectrophotometric and colorimetric results, please, correct.
  • The Authors performed a serious work measuring a series of physical properties of both CNO and UBO. However, it is not clear, what was the reason for measuring all these parameters, as they do not seem to be relevant in terms of the main objective of the work, which is exploration of the antioxidant properties and polyphenol content. Please, give a more substantiated justification of these measurements. Also, please remove redundant parts, like description of hue and chroma (L169-173).
  • L190-195: are the differences described significant? 
  • Regarding metal content, it is not clear, whether the differences between UBO and CNO arise from the lichen itself or from any contamination resulting from the environment, from harvesting or preparation of the extract. Did the Authors check metal content of the lichen itself? Did they collect more pieces of lichens from different spots? If not, the differences measured are not conclusive, which should be reflected in the text of the paper.
  • L352:  UAC = 915.60 ± 1.80 μg/g UBO, L356-357: "total polyphenols 
    content (TPC) values calculated were 32.382 ± 0.174 µg PyE/g in UBO". Please, explain the apparent contradiction!
  • L373: Please, give a reason for the lower correlation between AA and TPC in the case of CNO.

Reviewer 6 Report

Popovici et al. presented the characterization of  Usnea barbata extract in Canola Oil. The article is well-organized, readable and clearly prepared. Only the statement in lines 145-147 seems too far-fetched in the absence of in vitro and in vivo studies. This paper is suitable to be published in this journal.